# *Shigella* hijacks the exocyst to cluster macropinosomes for efficient vacuolar escape

Yuen-Yan Chang[1], Virginie Stévenin[1], Magalie Duchateau[2], Quentin Giai Gianetto[2,3], Veronique Hourdel[2], Cristina Dias Rodrigues[1], Mariette Matondo[2], Norbert Reiling[4,5], Jost Enninga[1] *

**1** Dynamics of Host-Pathogen Interactions Unit and CNRS UMR3691, Institut Pasteur, Paris, France, **2** Mass Spectrometry for Biology Unit, Proteomics Platform, Institut Pasteur, USR CNRS, Paris, France, **3** Hub Bioinformatics et Biostatistics, Computational Biology Department, USR CNRS, Institut Pasteur, Paris, France, **4** Microbial Interface Biology, Research Center Borstel, Leibniz Lung Center, Borstel, Germany, **5** German Center for Infection Research (DZIF), Partner site Hamburg-Lübeck-Borstel, Borstel, Germany

* jost.enninga@pasteur.fr

**Data Availability Statement:** All relevant data are within the manuscript and its Supporting Information files.

## Abstract

*Shigella flexneri* invades host cells by entering within a bacteria-containing vacuole (BCV). In order to establish its niche in the host cytosol, the bacterium ruptures its BCV. Contacts between *S. flexneri* BCV and infection-associated macropinosomes (IAMs) formed *in situ* have been reported to enhance BCV disintegration. The mechanism underlying *S. flexneri* vacuolar escape remains however obscure. To decipher the molecular mechanism priming the communication between the IAMs and *S. flexneri* BCV, we performed mass spectrometry-based analysis of the magnetically purified IAMs from *S. flexneri*-infected cells. While proteins involved in host recycling and exocytic pathways were significantly enriched at the IAMs, we demonstrate more precisely that the *S. flexneri* type III effector protein IpgD mediates the recruitment of the exocyst to the IAMs through the Rab8/Rab11 pathway. This recruitment results in IAM clustering around *S. flexneri* BCV. More importantly, we reveal that IAM clustering subsequently facilitates an IAM-mediated unwrapping of the ruptured vacuole membranes from *S. flexneri*, enabling the naked bacterium to be ready for intercellular spread via actin-based motility. Taken together, our work untangles the molecular cascade of *S. flexneri*-driven host trafficking subversion at IAMs to develop its cytosolic lifestyle, a crucial step *en route* for infection progression at cellular and tissue level.

## Author summary

*Shigella flexneri* is a clinically relevant bacterial pathogen that causes bacillary dysentery. It invades the host cell by injecting a repository of bacterial effectors through its type III secretion system. Upon its entry into host cell, *S. flexneri* resides shortly in a bacteria-containing vacuole (BCV), which is rapidly ruptured for the cytosolic propagation and infection progression of the pathogen. Infection-associated macropinosomes (IAMs) formed *in situ* during *S. flexneri* entry are found to promote the efficient vacuolar escape of *S. flexneri* with an unclear mechanism. We present here the molecular players involving in the

**Funding:** This work is supported by a fellowship from the Fondation pour la Recherche Medicale (FRM-SPF20160936275) to Y-Y.C. and by grants from the ERC (EndoSubvert) and the ANR (StopBugEntry and AutoHostPath). J.E. is a member of the LabEx consortium IBEID and MilieuInterieur. The funders had no role in study design, data collection and analysis, decision to publish, or preparation of the manuscript.

**Competing interests:** The authors have declared that no competing interests exist.

BCV-IAM interactions obtained by proteomic analysis of magnetically-purified IAMs. We decipher the successive steps of *S. flexneri* BCV escape, pinpointing the bacterial effector-mediated hijacking of host trafficking pathways to promote the BCV disintegration and displacement of BCV membranes from the bacteria. This study sheds light on the mechanism by which bacterial pathogens modulate BCV vacuolar integrity through interaction with infection-induced vesicle compartments for the establishment of their intracellular replicative niches and pathogenicities.

## Introduction

During bacterial invasion of eukaryotic cells, bacteria were internalized inside a bacteria-containing vacuole (BCV), which is then manipulated to establish the intracellular replicative niche of the pathogens. Upon entry, bacterial pathogens either replicate in spacious BCVs (*Yersinia*, *Chlamydia*), or they escape from their vacuoles to propagate inside the host cytoplasm (*Listeria*, *Shigella*) [1, 2]. Versatile strategies are thus deployed by bacterial pathogens to modulate the BCV stability in order to evade the host immune response and adapt to the hostile environment of the infected host [3–6]. One such strategy is to hijack host membrane trafficking pathways. These pathways are tightly regulated by the small GTPases of the Rab family to allow the recycling of endocytosed proteins and lipids back to the plasma membrane [7]. Rab GTPases coordinate an inventory of molecular players, named effectors [7, 8]. These effectors include the tethering factors and the motors of the cytoskeletal networks for vesicular transport [9]. Recycling vesicles are tethered to the plasma membrane by tethering factors, which enhances the subsequent membrane fusions mediated by soluble N-ethylmaleimide-sensitive factor attachment protein receptors (SNAREs) [7, 10].

Vacuolar escape signifies the prominent step of the infection cycle of a cytosolic bacterium. Studies on the BCV destabilization are, however, particularly challenging as the initial BCV lysis occurs in minutes after bacterial internalization in order to avoid the vacuolar acidification and lysosomal degradation [11, 12]. Moreover, studies of the intracellular dynamics of bacteria and host proteins are constrained due to a lack of robust methodology to identify molecular players involved in these transient interactions [13]. Details on the disintegration of the vacuolar membrane by bacterial pathogens are thus poorly understood.

*Shigella flexneri* is an enterobacterium that causes bacillary dysentery, which triggers its entry into non-phagocytic epithelia via the injection of effector proteins of the type III secretory system (T3SS) into the host cells [14, 15]. Membrane ruffling is then induced, in conjunction with the formation of infection-associated macropinosomes (IAMs) [16]. Two distinct compartments are thus formed: the tight bacteria-containing vacuole (BCV), within which *S. flexneri* resides, and the surrounding IAMs [16]. Being a professional cytosolic bacterial pathogen, *S. flexneri* promptly escapes from its vacuole and moves via an actin comet tail for intracellular motility and subsequent spreading to the neighboring cell [11, 17]. A number of reports have suggested how *S. flexneri* dismantles its BCV to gain cytosolic access [16–18]. On one hand, several *S. flexneri* T3SS effectors being injected into the cytosol of the infected host are validated to directly promote BCV lysis [18]. On the other hand, our intriguing observation that IAMs come into contact with the BCV just before vacuolar rupture implies the BCV destabilization upon interaction with the IAMs formed *in situ* [16]. Moreover, *S. flexneri* perturbs Rab5 and Rab11 trafficking on the IAMs in the infected host via its T3SS effector IpgD for efficient vacuolar escape, implying a functional implication of the host trafficking pathways in

this process [16, 17]. Nevertheless, the means by which *S. flexneri* subverts IAMs to escape from the ruptured BCV and promote intra- and intercellular motility have remained elusive.

Here, we utilize a large-scale proteomic approach, previously established in our laboratory, to magnetically isolate IAMs from infected cells and identify host factors involved in the cross-talk between IAMs and the *S. flexneri* BCV. We reveal two successive steps involved in the vacuolar escape of *S. flexneri*: the initial breakage of the BCV membrane that is then followed by the fragmentation of the damaged vacuole with its membrane remnants being carried away from the bacterium. Using time-lapse and automated quantitative microscopic analyses, we elucidate the novel role of proteins related to host recycling and exocytic pathways, in particular the exocyst, in an initial step of IAM clustering around *S. flexneri* BCV. We show that the clustered IAMs facilitate the unwrapping of ruptured BCV membranes from the naked bacterium, effectively freeing the pathogen in the cytosol. Collectively, our findings depict a framework of pathogen subversion of host exocytic trafficking, leading to interactions between BCV and IAMs that promote intracellular bacterial progression to the cytosolic stage of the infection.

## Results

### The proteome of infection-associated macropinosomes

Two distinct compartments, BCV and surrounding IAMs, are simultaneously formed in close vicinity upon *S. flexneri* entry to the host cell. We have previously demonstrated that the communication between these two compartments leads to the destabilization of *S. flexneri* BCV. To identify proteins involved in the BCV-IAM interactions, we extracted IAMs by magnetic purification, followed by an unbiased global proteomic analysis. Our approach to isolate IAMs is based on the previously described cell fractionation methodologies employing a strong magnet [6, 19]. We loaded the extracellular medium of HeLa cells with the dextran-coated magnetic beads (diameter of 100 nm) and subjected the cells to *S. flexneri* infection for 30 min. The beads were taken up within IAMs upon the closure of the membrane ruffles generated during *S. flexneri* internalization into host cells. A moderate multiplicity-of-infection (MOI) was chosen to reduce the probability of *S. flexneri* uptake inside macropinosomes filled with the magnetic beads. Afterwards, samples were fixed and immuno-stained for Rab5A that labels macropinosomes [20]. Confocal microscopic analysis showed that the magnetic beads were enriched at the *S. flexneri* infection focus colocalizing within Rab5A-positive vesicles (S1A Fig), validating the successful internalization of magnetic beads inside *S. flexneri* IAMs.

To analyze the protein composition of the *S. flexneri* IAMs, a large number of HeLa cells ($>10^8$ cells) were incubated with *S. flexneri* and magnetic beads. The infected cells were washed off extracellular bacteria and the beads, harvested and then lysed. The homogenate containing the vesicular compartments of the infected cells was subjected to magnetic extraction to isolate IAMs containing the magnetic beads (as illustrated in Fig 1A). As control, we incubated HeLa cells with the magnetic beads in the absence of the bacteria, to identify non-specific bead-protein binding (Fig 1A). We obtained the magnetic fractions (M fraction) and non-magnetic fractions (NM fraction) for both infected (INF) and control samples (Ctrl). Examination of the magnetic (INF-M) and non-magnetic (INF-NM) fractions of infected samples under a confocal microscope showed that the magnetic beads were exclusively present in the INF-M fraction compared to the INF-NM fraction, demonstrating the successful extraction of IAMs in the INF-M fraction (Fig 1B).

All isolated fractions were analyzed by mass spectrometry. Using a quantitative proteomic analysis, we compared the enrichment of proteins between (i) the M fraction and NM fraction of infected samples (INF-M *vs* INF-NM) and between (ii) the M fraction of infected samples

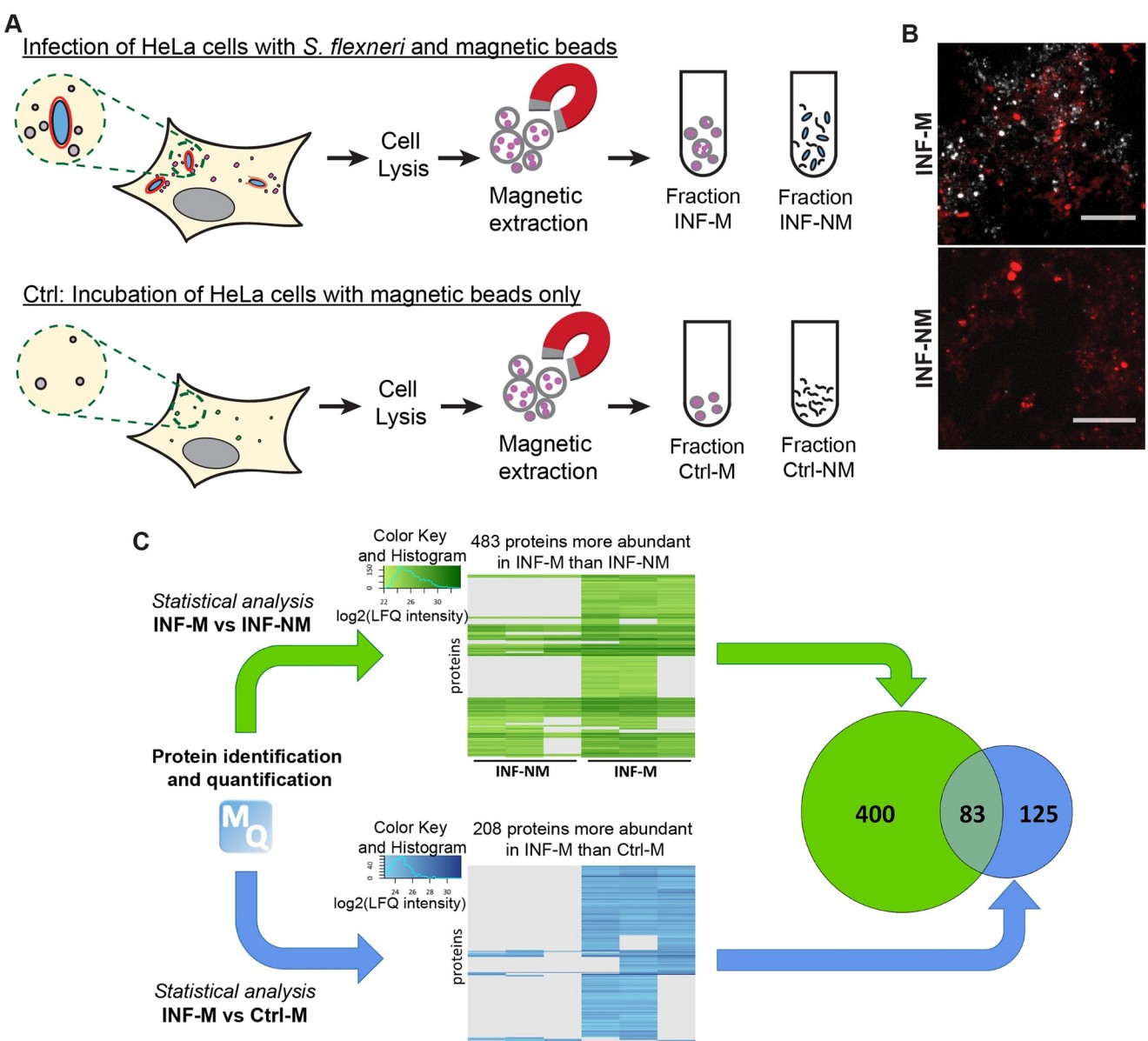

**Fig 1. Proteomic analysis of infection-associated macropinosomes (IAMs) during *S. flexneri* invasion.** (A) Schematic illustration of the magnetic purification of IAMs at *S. flexneri* infection. HeLa cells were incubated with magnetic beads while infecting with *S. flexneri*. HeLa cells incubated with the magnetic beads only served as control. Afterwards, cells were lysed and vesicles loaded with the magnetic beads were isolated and were subjected to proteomic analysis. (B) The magnetic (M) and non-magnetic (NM) fractions of the *S. flexneri*-infected samples (INF) were examined by confocal microscopy. Magnetic beads were shown in grey while membrane was stained by FM2-10 (red). Scale bar is 5 μm. (C) Comparative analyses of i) the M and the NM fractions of the infected sample (INF-M *vs* INF-NM, green) and ii) the M fractions between the infected (INF) and the HeLa control (Ctrl) samples (INF-M *vs* Ctrl-M, blue) after protein identification and quantification by mass spectrometry. The numbers in the Venn diagram indicate the numbers of identified proteins with at least two-fold enrichment in the respective fractions in the two comparisons. (D) List of the enrichment of some of the host proteins on *S. flexneri* IAMs according to their functions and their enrichment in the respective fractions.

and M fraction of uninfected control (INF-M *vs* Ctrl-M) to identify the host factors specifically enriched on the *S. flexneri* IAMs relative to the endogenous endosomes (Fig 1C; S1 Appendix). We found that 483 proteins were at least 2-fold more significantly enriched in the INF-M compared to INF-NM fractions (Fig 1C, green). On the other hand, 208 proteins were found to be at least 2-fold enriched in the INF-M compared to Ctrl-M fractions (Fig 1C, blue). Our comparative analyses showed 83 proteins are more abundant in INF-M than in the other two fractions (Fig 1C, Venn diagram). Analysis by the Database for Annotation, Visualization and Integrated Discovery (DAVID) showed that "microtubule-based transport" and "vesicular trafficking" were among the annotated clusters with the highest enrichment scores (S1B Fig). In order to decipher the factors that mediate the interaction between the *S. flexneri* BCV and the surrounding IAMs, we focused on proteins involved in vesicular trafficking in this study. Among those, we noted a significant enrichment (by applying a moderated t-test and imposing an enrichment at least twice higher in INF-M than in the other fractions, i.e. $\log_2 > 1$) of proteins involved in the recycling and exocytic pathway at IAMs: in particular the different subunits of the exocyst complex (Sec3, Sec5, Sec8, Sec15), together with the regulatory GTPase RalA and Rab8A (Fig 1D and S4 and S5 Tables).

## Transient recruitment of the exocyst to IAMs and the *S. flexneri* BCV

The exocyst is an octameric complex, consisting of subunits Sec3, Sec5, Sec6, Sec8, Sec10, Sec15, Exo70 and Exo84 [10]. Several structural analyses point towards an organization of the exocyst in two subcomplexes: Sec3, Sec5, Sec6 and Sec8 constitute one subcomplex (named "subcomplex 1" hereafter) and Sec10, Sec15, Exo70 and Exo84 constitute the other one (named "subcomplex 2" hereafter) [21–23]. The exocyst is a well-characterized tethering complex that facilitates vesicular interaction by bringing the interacting vesicles in close proximity [9]. In our proteomic analysis, four of the eight exocyst subunits (Sec3, Sec5, Sec8, Sec15) were significantly enriched on the extracted *S. flexneri* IAMs. Previous studies have shown that the interacting partner of the exocyst, Rab11, is recruited to *S. flexneri* IAMs through the *S. flexneri* T3SS effector IpgD [16, 17]. Of note, we detected Rab11 in the *S. flexneri* IAM proteome but not being classified as significantly enriched (i.e. < 2-fold enrichment). This is possibly due to the relative abundance of Rab11 in other cellular compartments.

We first confirmed the recruitment of the endogenous exocyst subunit Sec3, Sec5 and Exo70 to the bacterial entry foci by confocal microscope (S2A–S2C Fig). We validated that endogenous Sec3, Sec5 and Exo70 localize, together with Rab11A, on some IAMs at the close proximity to *S. flexneri* (S2A–S2C Fig). We then monitored the dynamic recruitment of GFP-tagged exocyst subunits to the *S. flexneri* BCV and the IAMs by live-cell imaging. In order to monitor the step of vacuolar rupture, Galectin-3-mOrange, which localizes to the inner leaflet of the BCV upon damage, was used as a marker of membrane damage [24, 25]. Time-lapse analysis at high spatiotemporal resolution confirmed Sec3 (Fig 2A; S1 Movie), Sec5 (Fig 2B; S2 Movie), Sec8 (S2D Fig), Sec15 (S2E Fig) and Exo70 (Fig 2C; S3 Movie) recruitment to the IAMs around invading *S. flexneri* (as indicated by the blue arrowheads in the figures). We

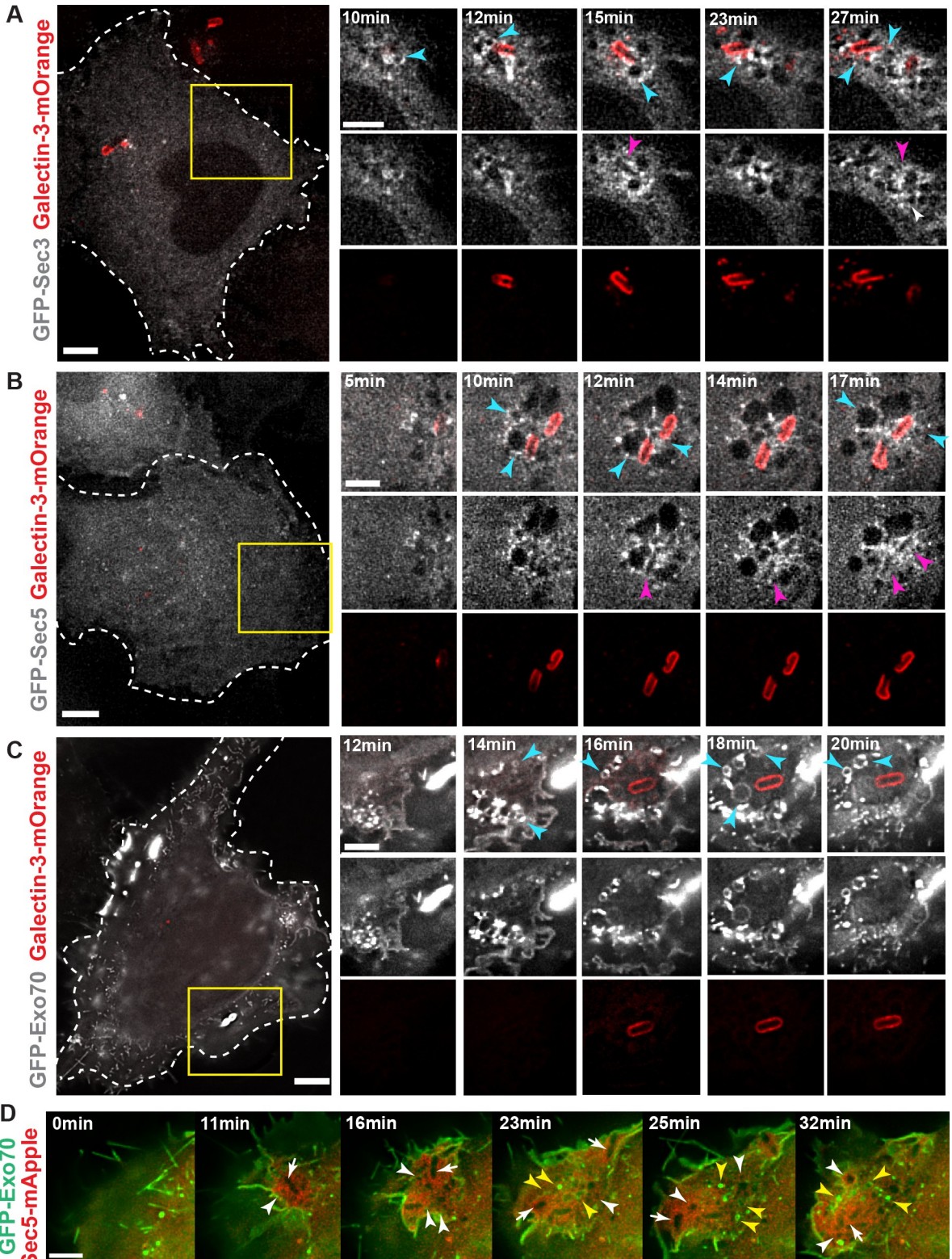

**Fig 2. Recruitment of the exocyst subunits during *S. flexneri* invasion.** Time-lapse microscopic analysis of recruitment of the exocyst subunits (grey) (A) Sec3, (B) Sec5 and (C) Exo70 in transfected HeLa cells. Galectin3-mOrange (red) was used as a marker of vacuolar rupture. Images were recorded every minute and z-projections of representative entry sites are shown. The blue arrowheads indicate the IAMs enriched with the respective exocyst subunits. The pink arrowheads in A-B indicate the sites where the respective exocyst subunits

localized at close proximity to the BCV. Scale bars are 5 μm. (D) Analysis of Exo70 (green) and Sec5 (red) recruitment by time-lapse microscope indicated that Exo70 was present at the IAMs (indicated by the yellow arrowheads) while Sec5 was localized at the IAMs and BCV (indicated by the white arrowheads and white arrows, respectively). Scale bar is 5 μm.

noted that the recruitment of the exocyst subunits became apparent 2–4 min before the instant of vacuolar rupture and post-BCV rupture. Besides, we also found that Sec3 and Sec5, in particular Sec5, were bordering the *S. flexneri* BCV in all the observed cases (Fig 2B, pink arrowheads), whereas Exo70 was exclusively present on the IAMs (Fig 2C). We further validated this observation by performing *S. flexneri* infection of HeLa cells co-expressing eGFP-Exo70 and mApple-Sec5. Exo70 was only found on the IAMs (Fig 2D, yellow arrows) in contrast to Sec5 that was present on both IAMs and BCV (Fig 2D, white arrowheads and white arrows, respectively). Considering that Sec5 and Exo70 each belong to the different exocyst subcomplexes that further assemble to tether interacting membranes, these findings suggest a role of the exocyst in the interactions between *S. flexneri* BCV and the surrounding IAMs.

## The exocyst is not necessary for the early invasion events of *S. flexneri*

*Shigella* and *Salmonella* employ a common trigger mechanism for internalization into host cells [26]. Upon membrane ruffling, *S. flexneri* is then enclosed in an actin-rich cage-like structure prior to BCV rupture [17]. A previous report suggests that the exocyst is involved in the exocytosis of vesicles at *Salmonella typhimurium* infection sites to provide membranes for membrane ruffling that facilitates bacterial entry [27]. To study if the exocyst is involved in facilitating *S. flexneri* entry to host cells, we examined the early events of *S. flexneri* invasion in control and Exo70-depleted HeLa cells in at least 135 infection foci per experimental condition (S3A Fig). In brief, actin-eGFP [28] was used to monitor the onset of actin ruffles upon incubation of *S. flexneri* with the cells (i.e. actin foci formation) and the appearance of the actin cage after the onset of ruffle (i.e. actin cage formation). Galectin-3-mOrange was also used as a BCV rupture marker to indicate the timing of BCV lysis after the onset of ruffle (i.e. BCV rupture time) [25]. Exo70 knock-down efficiency was confirmed by Western blot (S3B Fig). We found that the average time of actin foci formation in the Exo70-knockdown condition (24.1 ±1.1 min) was comparable to the control condition (24.7±1.0 min) (S3C Fig). Moreover, we did not find significant discrepancies in the timing of the subsequent actin cage formations (8.0±0.4 min in control and 7.4±0.3 min in Exo70-depleted condition) (S3D Fig) and *S. flexneri* BCV rupture (9.5±0.3 min in control and 9.4±0.3 min in Exo70-depleted condition) (S3E Fig) when Exo70 is absent in cells. These findings signify that the exocyst is dispensable neither for the membrane ruffling of *S. flexneri* nor the early steps following *S. flexneri* entry, which is in contrast to its reported role on enhancing *Salmonella* entry during bacterial invasion [27].

## The exocyst is involved in the clustering of IAMs around *S. flexneri*

We have previously demonstrated that reducing the number of IAMs formed *in situ* impedes the vacuolar escape of *S. flexneri* [16]. We confirmed in this work that the exocyst is recruited to the *S. flexneri* IAMs and in close proximity to its BCV (Fig 2). To study if the exocyst is functionally linked to the formation or the spatial organization of *S. flexneri* IAMs, we combined fluorescence microscopy and fluorescent dextran, a fluid phase marker, to label all IAMs at *S. flexneri* infection foci (marked by the fluorescent phalloidin) 30-min after challenging the HeLa cells with *S. flexneri*. More precisely, we quantify i) the availability and ii) the proximity of IAMs to invading *S. flexneri* in control and in exocyst-interrupted conditions. We found that the number (S4A Fig) and size of infection foci (S4B Fig) in Exo70-depleted cells were similar to the control, showing again that Exo70 is not critical for actin rearrangements during

*S. flexneri* ruffle formation. The average number of IAMs formed per *S. flexneri* in Exo70-depleted cells (10.2±1.3) was found to be comparable to control (8.8±1.0) (S4C Fig) with a similar trend in the size distribution of the IAM population (S4P Fig), suggesting that *S. flexneri*-induced IAM formation was not inhibited in Exo70-depleted conditions. We next investigated the spatial distribution of the IAMs in relation to the invading *S. flexneri* by quantifying the spread of the IAM population per invading bacterium (as illustrated in S4M Fig). Interestingly, we found that the IAMs in Exo70-depleted condition occupied a significantly larger area (38.3 ±2.0 μm$^2$) than in control conditions (27.5±1.3 μm$^2$) when normalized against the number of intracellular *S. flexneri*, indicating that IAMs are more dispersed when one of the exocyst sub-units is absent (Fig 3A).

The exocyst consists of two subcomplexes: Sec3, Sec5, Sec6 and Sec8 form subcomplex 1, whereas subcomplex 2 consists of Sec10, Sec15, Exo70 and Exo84 [21, 22]. Each subcomplex locates independently on the interacting membranes to assist the tethering of membranous organelles [22]. We hypothesized that the assembly of the exocyst is related to the distribution of IAMs during the event of *S. flexneri* vacuolar escape. We thus selected Sec5 from the subcomplex 1 to perform a similar analysis as we did on Exo70 from the subcomplex 2. First, we controlled the impact of Sec5 knock-down on the process of *S. flexneri* internalization after confirming the knock-down efficiency by quantitative PCR and Western blot (S4O Fig). As shown in S4D and S4E Fig, no difference in the number and size of infection foci were observed in Sec5-depleted cells, suggesting that the knock-down of Sec5 exerts insignificant effects on actin rearrangement at *S. flexneri* entry foci. In contrast, similar to the Exo70 knock-down results, we found that the area occupied by IAMs in Sec5-depleted cells (30.8±2.2 μm$^2$ per *S. flexneri*) was significantly larger than that in control cells (21.2±1.5 μm$^2$ per *S. flexneri*) (Fig 3B) despite a comparable number and sizes of IAMs formed in both conditions (S4F and S4P Fig).

The CorEx motif denotes the interface region of the exocyst subunits where the assembly of the complete complex occurs, which is also implicated in vesicular tethering [23]. To further investigated the mechanism of exocyst-dependent IAM clustering around *S. flexneri* BCVs, we constructed Sec5 and Exo70 mutants lacking the core exocyst assembly motif (named Sec5-ΔCorEx and Exo70-ΔCorEx hereafter) as reported in a previous functional study [23]. We confirmed that the expression of neither Sec5-ΔCorEx nor Exo70-ΔCorEx in HeLa cells exerted an impact on *S. flexneri* invasion (S4G, S4H and S4J, S4K Fig, respectively). *S. flexneri* was bordered by comparable numbers of IAMs in HeLa cells expressing Sec5-ΔCorEx and the wild-type protein (9.9±0.8 in Sec5-expressing cells and 11.0±1.2 in Sec5-ΔCorEx-expressing cells) (S4I Fig). Similarly, we observed a comparable number of IAMs formed in Exo70- and Exo70-ΔCorEx-expressing cells (11.6±1.0 in Exo70-expressing cells and 12.8±1.0 IAMs in Exo70-ΔCorEx-expressing cells), confirming again that the exocyst assembly is not essential for the formation of IAMs (S4L Fig). In contrast, the area of the spread of IAMs normalized per *S. flexneri* in Sec5-ΔCorEx-expressing cells (37.1±3.7 μm$^2$) is larger than that in Sec5-expressing HeLa cells (27.3±1.7 μm$^2$) (Fig 3C). We also observed a more scattered distribution of the IAMs (47.3 ±3.8 μm$^2$ per *S. flexneri*) in Exo70-ΔCorEx-expressing cells than in Exo70-expressing cells (39.1 ±2.7 μm$^2$ per *S. flexneri*) (Fig 3D). Furthermore, we applied automated image analysis to study *S. flexneri* invasion sites in detail, where we visualized the spatial distribution of the dextran-filled IAMs in 3D (S4N Fig). We observed a similar trend of wider ranges of distances between individual IAMs and *S. flexneri* in the conditions of knock-down of Exo70 and Sec5 (Fig 3E and 3F) or expression of the mutant proteins of the exocyst (Sec5-ΔCorEx and Exo70-ΔCorEx) (Fig 3G and 3H) compared to their respective controls. Therefore, all of the above results verify the notion that the exocyst complex and potentially the assembly of the fully functional complex is involved in the IAM clustering in close proximity to intracellular *S. flexneri*.

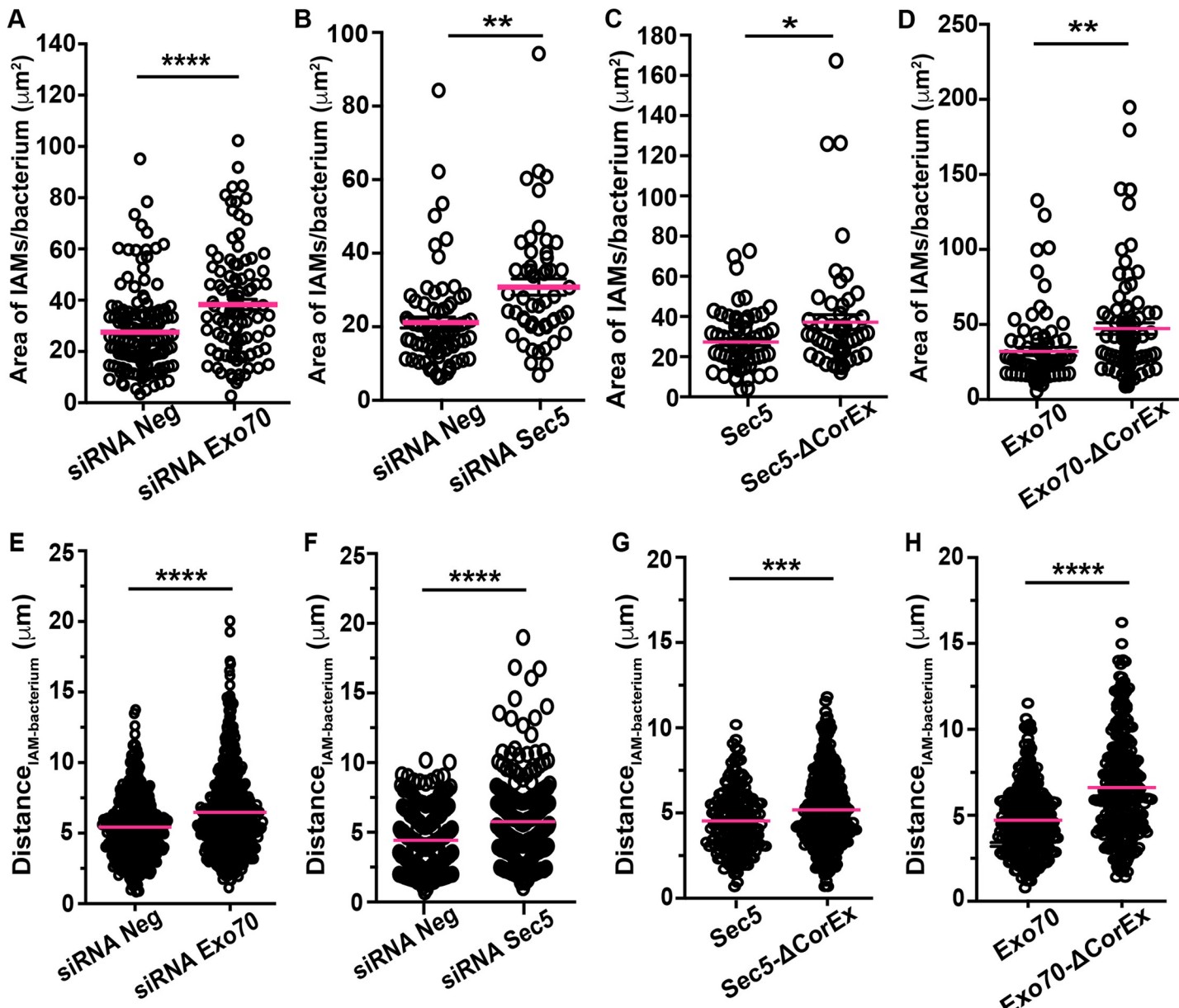

**Fig 3. The exocyst is required for tethering IAMs to the *S. flexneri*-containing vacuole.** *S. flexneri* infection of HeLa cells was performed in different conditions in the presence of fluorescent dextran, after which the areas occupied by the *S. flexneri* IAMs and the distances between individual IAMs and *S. flexneri* were quantified and compared in (A, E) cells subjected to RNA interference of non-targeting control (siRNA Neg) versus Exo70 depletion (siRNA Exo70), (B, F) cells subjected to RNA interference of non-targeting control (siRNA Neg) versus Sec5 depletion (siRNA Sec5) (C, G) cells expressing wild-type Sec5 versus cells expressing Sec5 lacking the core exocyst assembly motif (Sec5-ΔCorEx) and (D, H) cells expressing wild-type Exo70 versus cells expressing Exo70 lacking the core exocyst assembly motif (Exo70-ΔCorEx). At least a total of 55 infections foci were analyzed in triplicate experiments for each condition (n > 55). The bars (magenta) represent the mean and unpaired t-tests were carried out (*p<0.05; **p<0.01; ***p<0.001; ****p<0.0001).

## Displacement of membranes from *S. flexneri* after BCV lysis involves the exocyst on the IAMs

We analyzed at least 50 *S. flexneri* infection foci at 30-min post-infection in the control and in the exocyst subunits knock-down conditions (see above). We detected on average 2 intracellular bacteria per infection focus in a similar total number of infection foci in all conditions (S5A

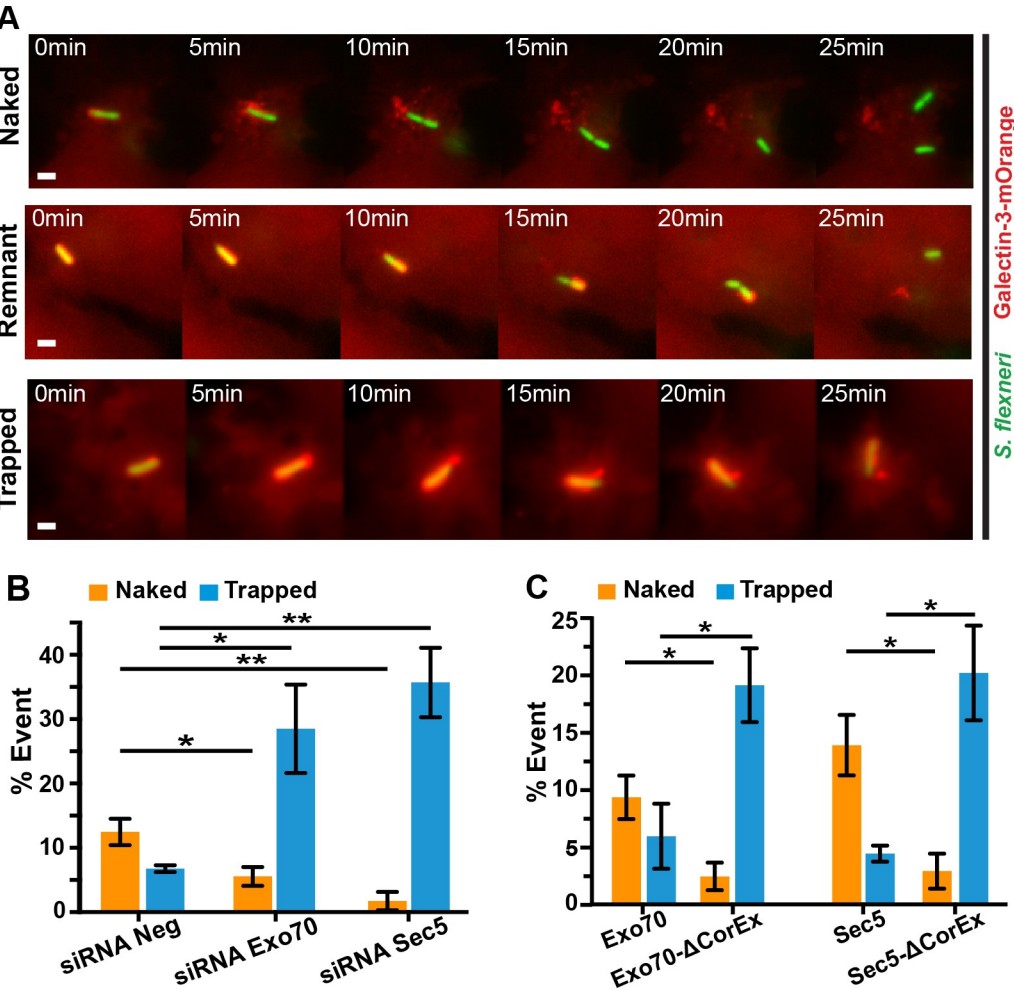

**Fig 4. The exocyst plays a role in the displacement of BCV membrane remnants from *S. flexneri*.** (A) Time-lapse microscopic acquisition of the fate of *S. flexneri* BCV was analyzed. Images were recorded every minute and the z-projections of representative infection foci were shown. *S. flexneri* expressing eGFP was employed (green), while ruptured BCV was indicated by Galectin-3-mOrange (red). Scale bars are 2 μm. Representative images of *S. flexneri* moving free of BCV membrane ("Naked", upper panel), *S. flexneri* moving with some BCV remnants ("Remnant", middle panel) and *S. flexneri* moving with the damaged vacuole ("Trapped", bottom panel). Analysis of the fates of individual *S. flexneri* and their BCVs with reference to the observations in (A) in different conditions including (B) RNA interference of non-targeting control (siRNA Neg) versus Exo70 depletion (siRNA Exo70) or Sec5 depletion (siRNA Sec5) and (C) expression of Exo70 or Sec5 versus their mutants lacking the core exocyst assembly motif (Exo70-ΔCorEx and Sec5-ΔCorEx, respectively). Data are shown as mean ± SEM in triplicates of each condition ($^{*}p<0.05$; $^{**}p<0.01$).

Fig), indicating that *S. flexneri* invades as efficiently in the exocyst subunits knock-down cells as in the control. We continued to study the effect of the exocyst on intracellular proliferation of *S. flexneri* at later hour post-infection by a gentamicin protection assay when Sec5 or Exo70 were depleted by RNA interference. We found a significant reduction in the bacterial load in both Sec5- or Exo70-depleted cells at 2 hr-post infection when compared to the control, suggesting that the exocyst is possibly involved in the progression of *S. flexneri* invasion after BCV rupture (S5B Fig). By time-lapse microscopy we observed instances where *S. flexneri* BCV membranes disintegrated in minutes after vacuolar rupture and these membranes were further displaced away from the bacterium. We qualified these bacteria as "naked", representing that *S. flexneri* that were free from BCV fragments in less than 20 min after vacuolar rupture (Fig 4A, top

panel–"Naked"; S4 Movie). Interestingly, in other cases, some BCV membrane remnants remained attached to *S. flexneri* (Fig 4A, middle panel–"Remnant"; S5 Movie) or because *S. flexneri* was still entrapped in the damaged vacuole even after 25 min post-BCV rupture (Fig 4A, bottom panel–"Trapped"). Furthermore, some bacteria were not able to leave the perforated BCV until the end of the acquired videos. We thus investigated in detail the role of the exocyst after the initial breakage of BCV using high temporal resolution imaging. Using Galectin-3-mOrange to label the BCV membrane remnants, we traced the fate of individual *S. flexneri* and the disassembly of their BCV membrane remnants. More precisely, we quantified the situations with clearly discernible phenotypes–i) "naked" *S. flexneri* and ii) *S. flexneri* that are "trapped" within a damaged BCV throughout the acquired videos. We observed about 12% of "naked" *S. flexneri*, whereas 6% of them were "trapped" in the damaged vacuoles in the scrambled knock-down control (Fig 4B). The depletion of exocyst subunits inverted these phenotypes, leading to only about 5% and 1% of "naked" *S. flexneri* in Exo70- or Sec5-depleted conditions, respectively (Fig 4B, "Naked"). In contrast, about 28% and 35% of *S. flexneri* being trapped within a damaged vacuole in Exo70- and Sec5-depleted cells, respectively (Fig 4B, "Trapped").

We further elucidate the role of the exocyst on the release of *S. flexneri* from its damaged BCV by monitoring individual *S. flexneri* and the surrounding BCVs in conditions of impaired exocyst assembly (*i.e.* cells expressing Exo70-ΔCorEx or Sec5-ΔCorEx). We observed a three-fold increase in proportions of the "trapped" *S. flexneri* in cells expressing the mutant proteins of the exocyst subunits (19% in Exo70-ΔCorEx-expressing cells and 20% in Sec5-ΔCorEx-expressing cells), compared to the respective wild-type proteins (Fig 4C). In contrast, the proportion of *S. flexneri* that moved naked decreases by 3.5-fold in cells expressing Exo70-ΔCorEx or by 5-fold in cells expressing Sec5-ΔCorEx, compared to cells expressing the respective wild-type proteins (Fig 4C). These results pinpoint that the exocyst plays a role in promoting the efficient release of *S. flexneri* from the ruptured BCV membranes.

## Exocyst-assisted BCV disintegration facilitates the intracellular motility of *S. flexneri*

We found that disruption of the function of the exocyst leads to an increase in the likelihood of *S. flexneri* being confined within the damaged BCV. Considering the observation that the size of BCV membrane remnants attached on the moving bacteria are highly heterogeneous, we further investigated the ambiguous phenotypes of BCV membrane displacement by scrutinizing the vacuolar escape of individual bacterium after initial BCV membrane damage. Specifically, we analyzed the timing of complete BCV disassembly, which we determined as the time at which the BCV completely disintegrates and is dissociated from *S. flexneri* after initial galectin-3 binding (Fig 5A, black rectangle). We also assessed the actin-based motility of *S. flexneri* using actin-mOrange to determine the timing of actin tail formation, which is defined as the time interval between the appearance of the actin cage around the BCV before rupture and the appearance of the actin tail after BCV rupture (Fig 5A, orange rectangle). We confirmed no considerable differences in the timing of actin foci formation, actin cage formation and BCV rupture between the knock-down control and the Sec5-depleted cells (S5C–S5E Fig). In contrast, a significant delay in both the BCV fragmentation time (18.5±1.3 min in control and 23.9±1.5 min in Sec5-depleted cells) (Fig 5B) and actin tail formation time (20.8±1.4 min in control and 26.2±1.6 min in Sec5-depleted conditions) were observed (Fig 5C). Similar delays in BCV fragmentation (25.7±1.4 min) (Fig 5D) and actin tail formation (22.7±0.9 min) (Fig 5E) were found upon Exo70-depletion.

Similarly, we carefully investigated the fate of individual *S. flexneri* and its BCV in Sec5-ΔCorEx- and Exo70-ΔCorEx-expressing HeLa. In contrast to the negligible effects on *S.*

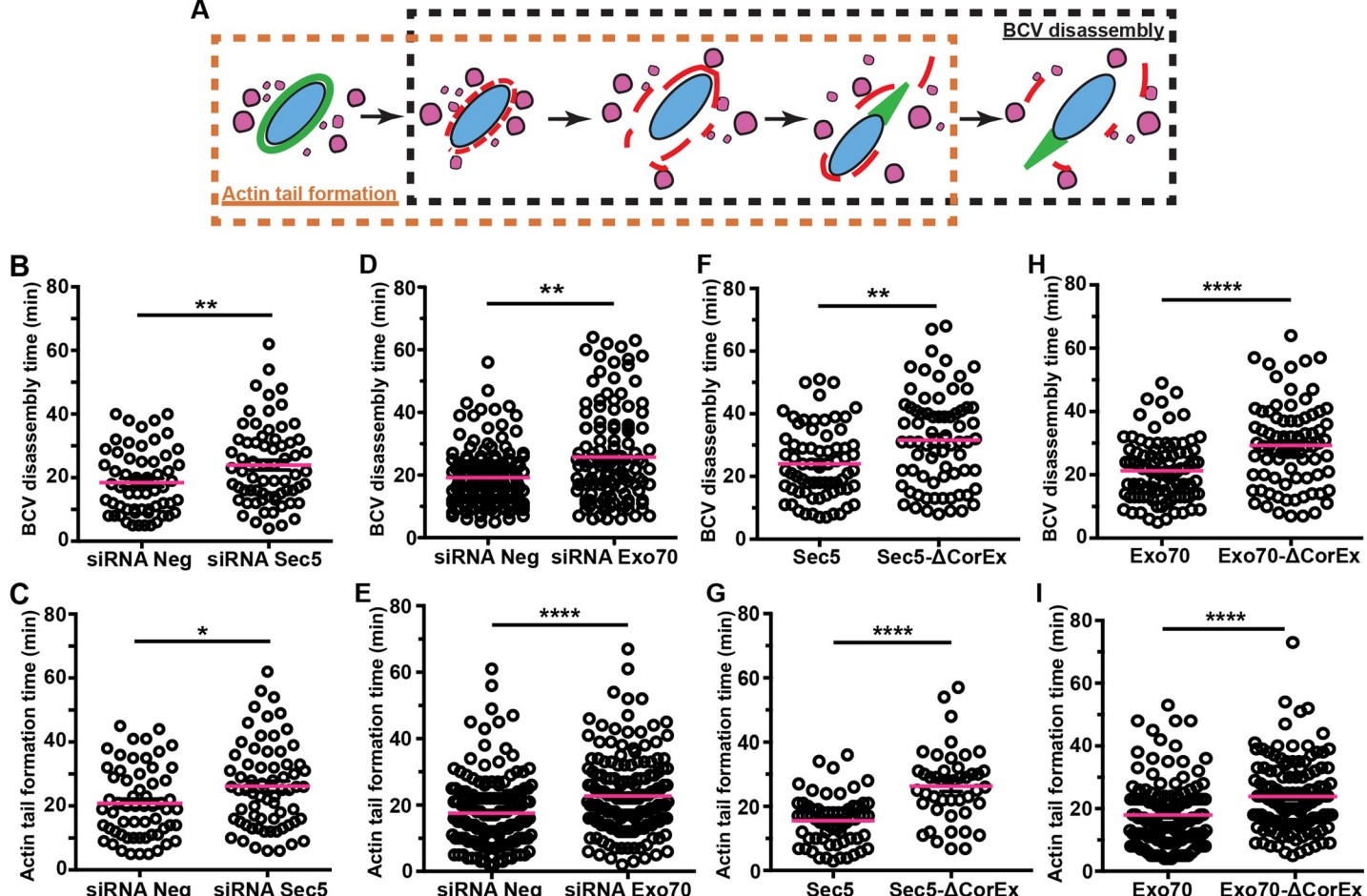

**Fig 5. The exocyst is involved in facilitating the efficient cytosolic escape of *S. flexneri*.** (A) Illustration of the study of the effect of the exocyst on *S. flexneri* invasion in relation to the time of disassembly of *S. flexneri* BCV (black rectangle) and the time of formation of actin tails (orange rectangle) in different conditions including (B-C) RNA interference of non-targeting control (siRNA Neg) versus Sec5 depletion (siRNA Sec5), (D-E) RNA interference of non-targeting control (siRNA Neg) versus Exo70 depletion (siRNA Exo70), (F-G) expression of wild-type Sec5 versus cells expressing Sec5 lacking the core exocyst assembly motif (Sec5-ΔCorEx) and (H-I) expression of wild-type Exo70 versus cells expressing Exo70 lacking the core exocyst assembly motif (Exo70-ΔCorEx). Individual BCVs and actin tails were analyzed in triplicates for each condition (n > 45). The bars (magenta) represent the mean and unpaired t-tests were carried out (*p<0.05; **p<0.01; ****p<0.0001).

*flexneri* entry (S4F–S4K Fig), we found that both BCV disassembly time (Fig 5F and Fig 5H) and actin tail formation time (Fig 5G and Fig 5I) were significantly delayed in Sec5-ΔCorEx- and Exo70-ΔCorEx-expressing HeLa, respectively, compared to the control conditions.

Our results implicate the functional role of the exocyst during *S. flexneri* infection. To further pinpoint the unique role of the exocyst in clustering *S. flexneri* IAMs for its BCV fragmentation, we carried out a similar microscopic analysis in the presence of Endosidin-2 (ES2), a reported specific chemical inhibitor of Exo70 [29]. ES2 binds to Exo70 and inhibits the transport of recycled cargoes to the plasma membrane, leading to the accumulation of exocytic vesicles at the cell periphery [29]. We treated the cells with ES2 for 4 hours according to a previous report [29], followed by *S. flexneri* infection of the treated cells. We first verified that ES2 treatment showed negligible effect on *S. flexneri* entry to host cell. Inhibiting Exo70 by ES2 did not affect the timing of actin foci formation, actin cage formation and BCV rupture (S6A–S6C Fig). Besides, treatment with ES2 showed no differences in the size of the actin foci (S6D Fig) and the overall numbers of entry foci (S6E Fig) compared to the control. In contrast, we found

a more dispersed distribution of the IAMs around *S. flexneri* in ES2-treated cells (S6G Fig) despite the same extent of IAM formation between the ES2-treated and the DMSO control cells (S6F Fig). A wider range of distances between individual IAMs and *S. flexneri* was also observed using a 3D analysis of the infection foci, postulating the IAMs are sparsely located around the bacterium (S6H Fig). We also found a lower *S. flexneri* load in ES2-treated cells at 2 hr-post infection, suggesting an inhibition on bacterial growth when the function of Exo70 was interrupted by the specific inhibitor ES2 (S6I Fig). We then examined individual *S. flexneri* and their surrounding BCV upon the initial BCV rupture by time-lapse microscopy. Interestingly, we observed more than a 5-fold rise in proportions of *S. flexneri* confined within a perforated vacuole in ES2-treated cells, compared to the DMSO control condition (3% in DMSO control cells and 22% in the ES2-treated cells) (S6J Fig, "Trapped"). Our microscopic analyses on the population of *S. flexneri* that developed the cytosolic lifestyle (i.e. excluding those associated with an entire damaged BCV in the acquired videos) showed that it took a longer time for the BCVs to undergo complete fragmentation (20.0±1.2 min and 29.6±1.7 min in DMSO control and in ES2-treated condition, respectively) (S6K Fig). Moreover, *S. flexneri* exhibited a delay in the formation of actin tails (17.9±1.1 min in DMSO control and 29.6±1.7 min in ES2-treated condition) (S6L Fig). These results thus verified the involvement of the exocyst in the BCV fragmentation, which promotes *S. flexneri* escape from its vacuole.

Together, we validated that *S. flexneri* surrounded by more dispersed IAMs have a higher probability to be enclosed with "vacuolar" remnants, resulting in a slower escape from the ruptured BCV. This implicates a correlation of the exocyst-mediated IAMs clustering around *S. flexneri* and the movement of the BCV membrane remnants away from the bacterium. Our results demonstrate that *S. flexneri* subverts the exocyst at two successive infection steps. First, the exocyst facilitates IAM clustering around the invading *S. flexneri*, where exocyst assembly (or interactions between exocyst subcomplexes) is critical in enhancing BCV-IAM interactions. Second, the exocyst is involved in the disintegration of the BCV, allowing an efficient cytosolic *S. flexneri* escape via actin-based motility.

## Rab8A and Rab11A are involved in recruiting the exocyst to *S. flexneri* IAMs

Rab11 is recruited to the *S. flexneri* IAMs before BCV lysis [16, 17]. Besides, Rab8A was identified as significantly enriched on *S. flexneri* IAMs in this study (Fig 1D). As the trafficking and tethering of recycling cargoes to the plasma membrane by the exocyst is regulated by the Rab GTPases like Rab11 [30, 31], we examined if the exocyst recruitment to IAMs is dependent on Rab8A or Rab11A. We confirmed the recruitment of Rab8A to IAMs prior to BCV rupture by time-lapse microscopy (S7A Fig). It was also found that Sec5 colocalizes with the endogenous Rab8A on some IAMs at a close proximity to *S. flexneri* (S7B Fig). We then expressed mApple-Rab8A or mApple-Rab11A, together with GFP-Exo70, to obtain precise information on their individual recruitment times. We found that Exo70 localized on some Rab8A- (Fig 6A; S6 Movie) or Rab11A-positive IAMs (Fig 6B; S7 Movie) (indicated by the white arrowheads in Fig 6A and 6B). Furthermore, we examined the Exo70 localization in the presence of the dominant negative mutants of Rab8A, Rab8A-T22N, (Rab8A-DN) (Fig 6C) and Rab11A, Rab11A-S25N, (Rab11A-DN) (Fig 6D). Rab8A-T22N and Rab11A-S25N mimic the GDP-bound structure of the respective Rab proteins, disrupting the activity of the Rab GTPases. It was observed that the Exo70 localization on Rab8A- or Rab11A-positive IAMs drops dramatically from about 90% to 33.3±7.3% in cells expressing Rab8A-DN (Fig 6E) and to 28.7±4.3% in cells expressing Rab11A-DN (Fig 6F), implying that the recruitment of the exocyst does depend on GTPase activity of Rab8A and Rab11A.

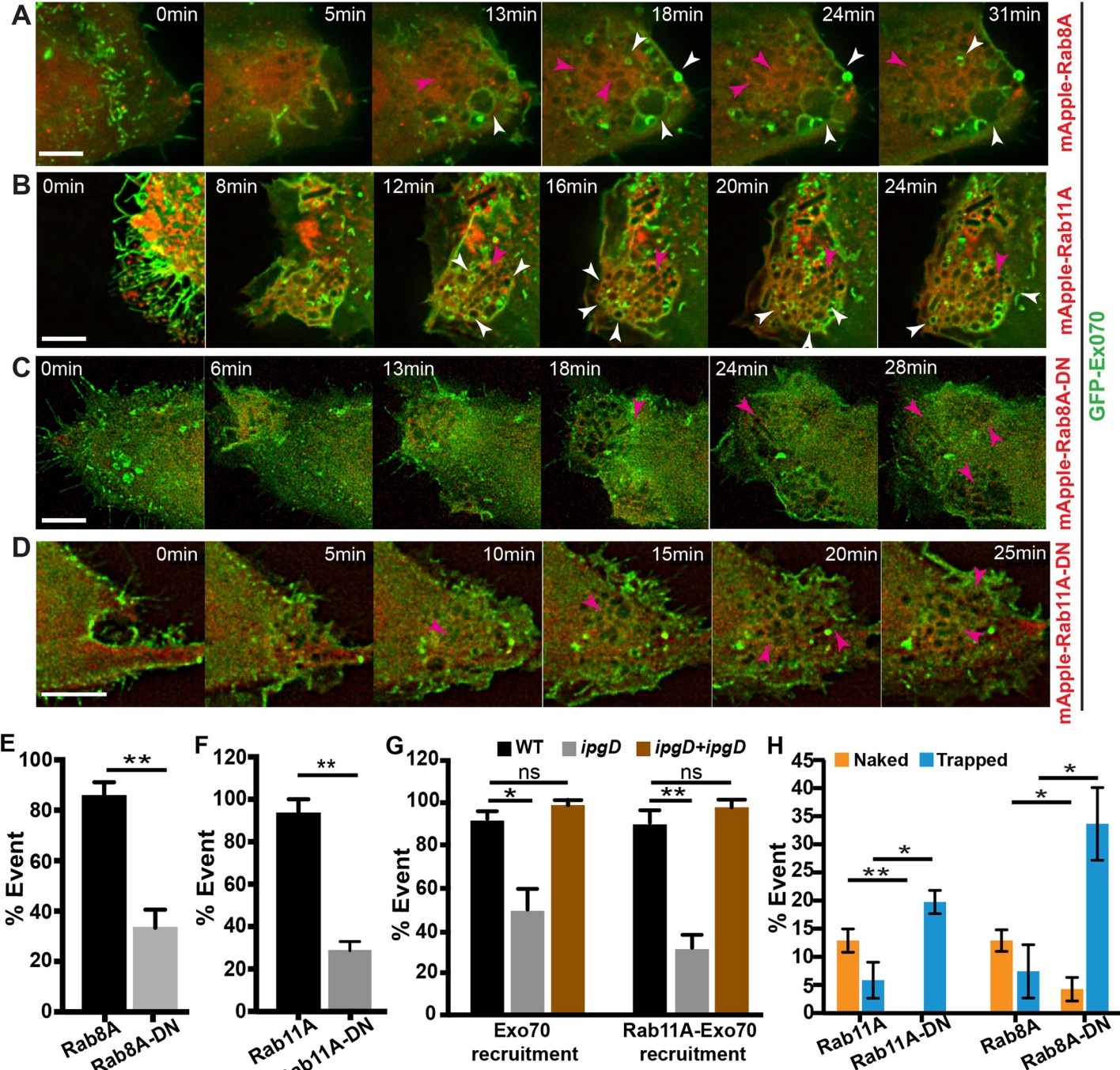

**Fig 6. Rab8A and/or Rab11A are involved in the recruitment of the exocyst at *S. flexneri* invasion site.** Time-lapse microscopic images of the recruitment of Exo70 (green) in the presence of (A) Rab8A, (B) Rab11A, (C) dominant-negative mutant of Rab8A, Rab8A-T22N (Rab8A-DN), and (D) dominant negative mutant of Rab11A, Rab11A-S25N (Rab11A-DN. Rab GTPases and their mutants are labelled in red. Images were captured every minute and the z-projections of representative infection foci were shown. The white arrowheads in A-B indicate the localization of Exo70 on some of the Rab8A- or Rab11A-positive IAMs. The magenta arrowheads in A-B indicate the localization of Exo70 on some of the Rab8A- or Rab11A-positive IAMs. The magenta arrowheads in A-D indicate the localization of *S. flexneri*. Scale bars are 5 μm. (E) Percentage of localization of Exo70 on a Rab8A- or Rab8A-DN-positive IAM in cells expressing the respective Rab proteins. At least 35 infection foci were analyzed in triplicates for each condition. Data are shown as mean ± SEM (**p<0.01). (F) Percentage of localization of Exo70 on a Rab11A- or Rab11A-DN-positive IAM in cells expressing the respective Rab proteins. At least 25 infection foci were analysed in triplicates for each condition. Data are shown as mean ± SEM (**p<0.01). (G) Percentage of localization of Exo70 on *S. flexneri* IAMs (left) and percentage of localization of Exo70 on a Rab11A-positive IAM in cells expressing Rab11A (right) when infected with the wild-type *S. flexneri* or the Δ*ipgD* mutant (*ipgD*) or the Δ*ipgD* mutant complemented with an *ipgD* expression plasmid (*ipgD*+ *ipgD*). At least 25 infection foci were analysed in triplicates for each condition. Data are shown as mean ± SEM in triplicates of each condition (*p<0.05; **p<0.01). (H) Analysis of the fates of individual *S. flexneri* BCVs with reference to the observations in Fig 4A in cells expressing Rab8A or Rab11A and their respective dominant-negative mutants (Rab8A-DN and Rab11A-DN). Data are shown as mean ± SEM in triplicates of each condition (*p<0.05; **p<0.01).

It has been previously reported that the *S. flexneri* T3SS effector IpgD is required for Rab11 recruitment to *S. flexneri* IAMs [17]. We found that recruitment of Exo70 to *S. flexneri* IAMs reduces from 93.0±3.6% to 50.0±9.8% when infected using the Δ*ipgD* mutant (Fig 6G), whereas the Exo70 localization on Rab11A-positive IAMs drops from 91.0±5.9% to 31.7±6.2% in the presence of the Δ*ipgD* mutant (Fig 6G). We further showed that complementation with a plasmid encoding the *ipgD* gene of the Δ*ipgD* mutant rescued the phenotypes completely (Fig 6G). These results imply that *S. flexneri* virulence effector IpgD mediates the interaction with Rab11, facilitating the recruitment of the exocyst to IAMs.

We further characterized the distribution of IAMs under the expression of inactive Rab8A and Rab11A. In brief, we infected HeLa cells expressing Rab8A-DN and Rab11A-DN with *S. flexneri* for 30 min in the presence of fluorescent dextran. We confirmed that the expression of Rab8A-DN and Rab11A-DN in cells has no impact on *S. flexneri* entry (S7C Fig). Besides, the number of IAMs formed in cells expressing the Rab mutants was not significantly different from control cells (S7D Fig). In contrast, we observed a significantly more scattered spatial distribution of the IAMs per intracellular *S. flexneri* in cells expressing the functionally inactive GTPase mutants (S7E–S7G Fig). Moreover, more than 3-fold increase in the proportion of *S. flexneri* being confined in damaged BCVs was observed in the presence of the non-functional Rab8A or Rab11A, compared to cells expressing the wild-type proteins (Fig 6H). These results were in agreement with the analyses of the disrupted exocyst in the conditions of RNA interference, expression of mutant proteins and use of Exo70 inhibitor. Our findings, together, corroborate that *S. flexneri* hijacks Rab11 and Rab8 to relocate the exocyst in order to cluster IAMs around *S. flexneri*, promoting the efficient disintegration of perforated BCV to facilitate cytosolic access and proliferation of the bacterium.

## Discussion

In order to survive and establish a replicative niche inside host cells, bacterial pathogens hijack the host membrane trafficking pathways to remodel or destabilize their BCVs. In this work, we unravel a novel role of the exocyst and provide a molecular cascade of host trafficking subversion mediated by the bacterial effector protein. The *S. flexneri* IAM proteome uncovers the enrichment of proteins in the recycling and exocytic pathways on *S. flexneri* IAMs. We confirmed the recruitment of exocyst subunits Sec3, Sec5, Sec8, Sec15 and Exo70 to *S. flexneri* IAMs. Our microscopic studies show that Sec5 was found on IAMs and in close proximity to the BCV whereas Exo70 was exclusively present on IAMs (Fig 7A), suggesting a potential role of the exocyst assembly for BCV-IAM communication. We further validate the Rab8- and Rab11-dependent recruitment of the exocyst to the IAMs (Fig 7A). This work thus reiterates our previous work [17] that this cascade is triggered via the bacterial effector IpgD (Fig 5G). The exocyst thus spatially organizes the BCV and IAMs to favor the unwrapping of *S. flexneri* from its damaged BCV and the progression of *S. flexneri* infection (Fig 7B). Taken together, our current study proposes a model of Rab8 and Rab11 being hijacked by *S. flexneri* to the IAMs, resulting in the exocyst-driven enhancement of IAM docking to the "plasma-membrane-like" BCV for efficient cytosolic *S. flexneri* escape from its vacuole.

We deployed a proteomic approach to elucidate the molecular players that modulate BCV-IAM interactions (Fig 1). The current work reaffirms the subversion of IAMs formed *in situ*, in addition to the previously proposed role in BCV lysis [16], for the efficient vacuolar escape of *S. flexneri* and progression to its intracellular cytosolic niche. Previously we noted that the IAMs formed during *S. typhimurium* infection can fuse with the early entry compartment of *S. typhimurium*. [6, 32]. Proteomic analysis of the IAMs during *S. typhimurium* infection also suggests specific SNARE proteins involved for the BCV-IAM fusion, implying a

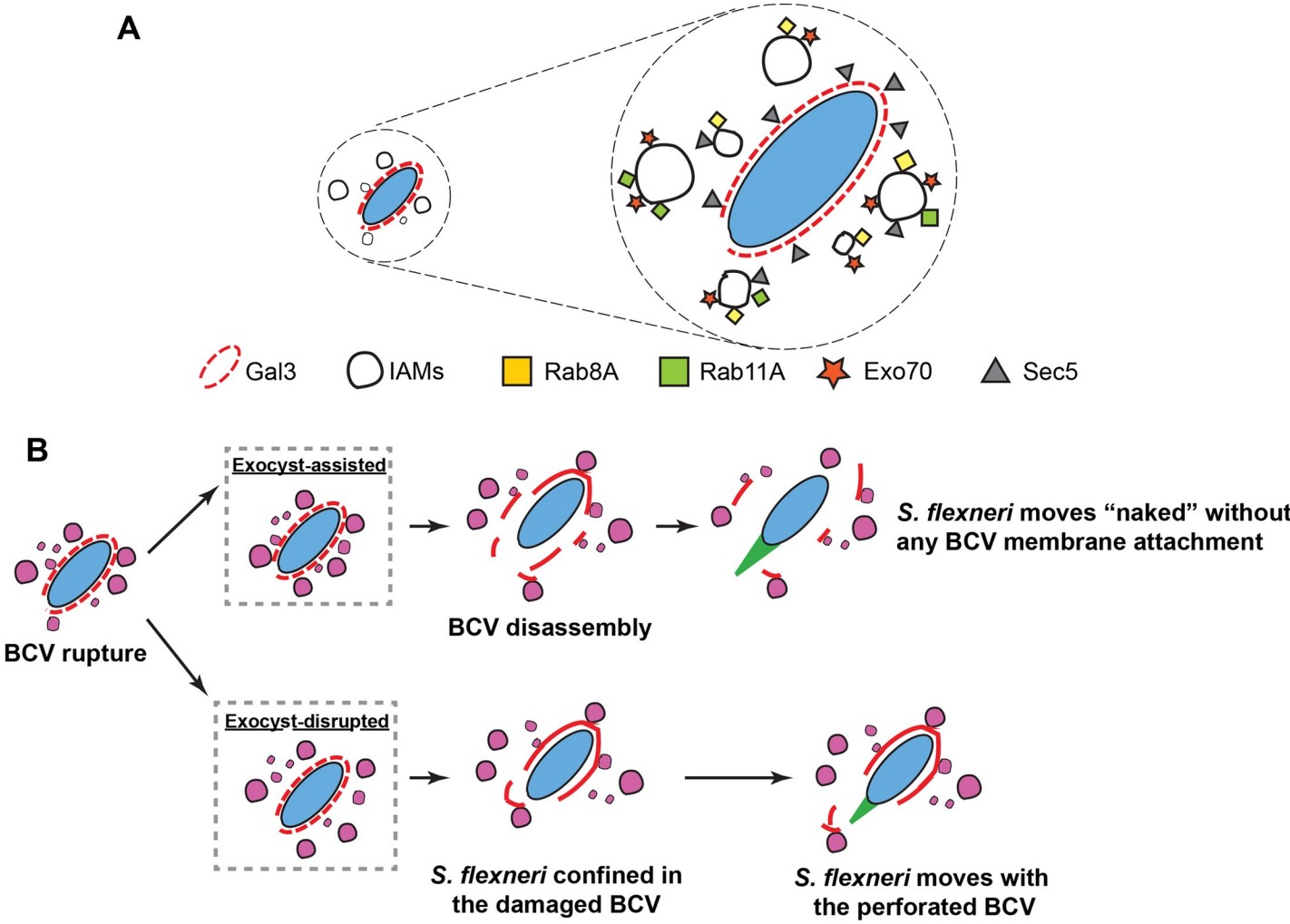

**Fig 7. Schematic illustration of the role of the exocyst on the efficient cytosolic escape of *S. flexneri*.** (A) Recruitment of Sec5 (triangle, grey) and Exo70 (star, orange) by Rab8A (square, yellow) and Rab11A (square, light green). Sec5 is present on the *S. flexneri* IAMs and BCV, whereas Exo70 is exclusively present on the IAMs. (B) Successive events after the BCV rupture of *S. flexneri*, in the exocyst-assisted or exocyst-disrupted conditions, are depicted. The exocyst assists in organizing IAMs in close proximity to the rupturing BCV, facilitating the BCV disintegration that leads to a high proportion of *S. flexneri* moving free of any attached BCV membrane remnants. In contrast, IAMs are more dispersed in exocyst-disrupted conditions (e.g. depletion of the exocyst subunits by siRNA or expression of mutants that interrupt the assembly of the exocyst complex, as demonstrated by Sec5-ΔCorEx and Exo70-ΔCorEx in this study, or in the presence of the specific inhibitor of Exo70). *S. flexneri* is observed to stay longer within the damaged BCV or moves with the attachment of the perforated BCV in these conditions.

functional implication of the IAMs in promoting the vacuolar lifestyle of the pathogen [6]. Interestingly, these SNARE proteins are not significantly enriched on *S. flexneri* IAMs. On the other hand, we found that the exocyst subunits are uniquely enriched on *S. flexneri* IAMs but not on *S. typhimurium* IAMs. Our proteomic analyses in this work and in our previous study [6] thus pinpoint that the molecular identities of the pathogen-induced macropinosomes are very unique to the pathogen invasion and other stimulus-induced macropinocytosis. Collectively, our findings highlight the under-studied role of IAMs in facilitating the establishment of the intracellular niches of bacterial pathogens. Considering that macropinocytosis is induced by different bacterial pathogens, such as *Chlamydia* and *Neisseria* [33, 34], it will thus be important to scrutinize whether other pathogens modulate of the vacuolar integrity through communication between vesicular compartments induced upon pathogen invasion.

Our current microscopy-based examination, together with others, validates the exploitation of the exocyst by intracellular bacterial pathogens for microbial pathogenesis [5, 27, 35–37]. Our analysis on the precise steps of bacterial invasion clearly reveals that the exocyst has a negligible effect on *S. flexneri* entry (S3 Fig), in contrast to the suggested entry enhancement by the exocyst in the infection of other bacteria like *Salmonella* and *Staphylococcus* [27, 37]. On the other hand, it has been reported that intracellular bacteria, such as *S. typhimurium*, reprogram the host exocytic processes to remodel their replicative vacuoles [38, 39]. Of note, the exocyst has been implicated in the maturation of *S. aureus*- or *L. pneumophila*-containing vacuoles attributable to fusion with vesicular compartments to encourage intravacuolar bacterial replication [5, 36]. Nonetheless, this does not appear to be the case in *S. flexneri* infection because we could not detect the fusions between *S. flexneri* BCV and the surrounding IAMs in the current work, as reported earlier [16]. Membrane fusion between vesicular compartments is a cooperative effort by a repository of proteins in membrane trafficking, including the Rab GTPase, tethering factor and SNAREs [7]. A recent report has suggested that some of the these factors involved in membrane fusion can possibly be modified by *S. flexneri* T3SS effector IcsB on *S. flexneri* BCV during bacterial infection [40]. One possibility is that the IcsB-mediated modifications of host trafficking factors hinders molecular interactions that drive membrane fusion, although more studies are needed to test this hypothesis. On the contrary, our quantitative microscopic analyses delineate precisely the novel role of exocyst in two intricate steps of *S. flexneri* vacuolar escape: clustering of IAMs around the bacteria and facilitating the disintegration of the damaged BCVs (Figs 3–4). We reason that the exocyst-assisted docking of IAMs on *S. flexneri* BCV may enhance the exchange of proteins or lipids between the BCV and the IAMs, promoting the efficient disassembly of *S. flexneri* BCV. Cytosolic *S. flexneri* may thus take advantages of the exocyst-assisted vacuolar destabilization to escape of targeting by guanylate-binding proteins (GBPs) [41–44], concomitant with evasion of the LC3-dependent autophagic signaling that is initiated by the damaged vacuole [45].

Some bacterial effectors have been reported to interact with the exocyst. The *Legionella* effector DrrA mediates the interaction with the exocyst via the formation of a DrrA-Rab1 complex, requiring the GEF domain of DrrA [5], whereas the *Salmonella* effector SipC directly interacts with the exocyst via its C-terminus [27]. In this work, we show that the T3SS effector IpgD participates in the recruitment of the exocyst to the IAMs through the interaction with Rab11A (Fig 6). Such an interaction could be attributed to the change in lipid composition of the Rab11 vesicles by the phosphatase activity of IpgD [17]. Our work thus expands the knowledge on the role of IpgD during *Shigella* invasion, in addition to its previously reported roles in releasing the membrane tension that facilitates cytoskeleton rearrangement for bacterial entry [46] and in promoting BCV lysis [17]. We note that we observed strongly inhibited, but not abolished, recruitment of the exocyst to IAMs in the presence of the IpgD mutant (Fig 6G), suggesting that other *S. flexneri* effectors may be involved in the interaction with the exocyst. One possible candidate of *S. flexneri* T3SS effectors is the above-mentioned IcsB that is suggested to modify Rab8A and Rab11A during *S. flexneri* infection [40]. IcsB could possibly play a role in interacting with the exocyst via interference with these Rab GTPases. Moreover, being a homologue of *Salmonella* SipC, *S. flexneri* T3SS effector IpaC shares some functional similarities with SipC in terms of the interaction with phospholipid membrane [47]. IpaC could possibly interact with the exocyst subunits on the IAMs directly through its C-terminus, similar to SipC. Nevertheless, it remains to be determined if *S. flexneri* deploys different strategies to achieve efficient recruitment of the exocyst to facilitate its vacuolar escape.

Cytosolic escape is crucial for the onset of secondary invasions of neighboring cells during *S. flexneri* infection, through intra- and subsequent intercellular spread [11]. Using the time-resolved measurements at the single bacterium level, we reveal and decipher the two distinct

steps involved in the unwrapping of *S. flexneri* from the surrounding BCV membrane: the observation of the initial BCV breakage, followed by the displacement of BCV fragments away from *S. flexneri* prior to cytosolic propagation via actin comet tails (as illustrated in Fig 7B). We observed that *S. flexneri* initiates restricted movements with attached BCV remnants or from within a perforated vacuole, in the absence of efficient BCV fragmentation (Fig 4). The increase in the probability to confine *S. flexneri* in a rupture BCV and the restrained bacterial mobility due to inefficient BCV fragmentation becomes predominant in conditions with abolished exocyst-assisted IAMs clustering, corroborating the correlation of IAM spatial distribution around the bacteria with the degree of fragmentation of the ruptured BCVs, implicating that BCV-IAM contacts influence the efficiency of which the pathogen escape from its vacuole. Although further investigations will help elucidating the exact mechanism by which *S. flexneri* benefits from the IAMs to disintegrate its BCV, we favors the postulation that *S. flexneri* subverts Rab11-positive IAMs as localized hubs to relay biomolecules like Rab11 effectors. Rab11 coordinates factors in the trafficking pathways for vesicle transport, including the exocyst and the molecular motors [30, 48, 49]. Given that the translocon complex of *S. flexneri* T3SS and the T3SS effectors is sufficient for *S. flexneri* BCV lysis [18, 47], the mechanical forces exerted by the surrounding Rab11-positive IAMs on the BCV may further boost the disintegration of the vacuole. Taken together, our current work provides, for the first time, to our knowledge, strong evidence of the subversion of host trafficking pathways on the IAMs by an invasive bacterial pathogen to boost its escape from the BCV. This work underscores the importance of the mechanistic understanding of the vacuole membrane dynamics of BCVs, potentially impeding the establishment of the intracellular niches of bacterial pathogens and their pathogenicity.

## Materials and methods

### Molecular cloning

Plasmids in this study were constructed by PCR unless otherwise stated. For details on the plasmids and the PCR primers, see S1 and S2 Table, respectively.

### Cell culture

HeLa cells were cultured in DMEM supplemented with 10% fetal bovine serum (FBS) at 37°C in the presence of 5% $CO_2$. Transfection of plasmids was performed using FuGENE Transfection Reagent (Promega) according to the manufacturers' protocol. ON-TARGETplus Smartpool siRNA for Exoc2 (L-017357-00-0005) and Exoc7 (L-021448-00-0005) were purchased from Dharmacon, GE Healthcare. Non-targeting siRNA ON-TARGETplus (D-001810-10-05) (Dharmacon, GE Healthcare) served as the siRNA control. siRNA transfection was performed using Lipofectamine RNAiMAX transfection reagent (Thermo Fisher) at a final concentration of 10 nM for 72 hr prior to infection. Protein knock-down efficiency was confirmed by Western blot.

### Bacterial strains and infection

*S. flexneri* serotype M90T strain was used in all experiments unless specified. The Δ*ipgD* mutant was kindly provided by John R. Rohde (Dalhousie University). Complementing the *ipgD* gene of the Δ*ipgD* mutant was done by transforming a pBAD18-*ipgD* plasmid into the Δ*ipgD* mutant, where the expression of IpgD was induced in the presence of 0.2% (w/v) L-arabinose during subculture [17]. All bacterial strains express the adhesin *afaI* unless specified. *S. flexneri* serotype M90T strain constitutively expressing the green fluorescent protein (eGFP) was established by transforming plasmid with *egfp* gene under the rpsM promoter (M90T-GFP).

M90T-GFP was used in time-lapse live imaging, which required poly-L-lysine treatment at room temperature condition for 15 min prior to the infection experiment as described before [50]. All bacteria strains were grown in TCS medium supplemented with 100 μg/mL ampicillin at 37 °C. On the day of infection, bacteria were subcultured in 1:100 dilution in TCS medium supplemented with 100 μg/mL ampicillin at 37 °C until an OD600 of ~0.5. Bacteria were harvested using centrifugation at 8000 *g* for 1 min and then washed once in EM buffer (25 mM HEPES, pH 7.3, 120 mM NaCl, 7 mM KCl, 1.8 mM $CaCl_2$, 0.8 mM $MgCl_2$, 5 mM glucose). Bacteria were resuspended in EM buffer and diluted to MOI 20 for fixed experiment or MOI 50 for magnetic purification and for live experiment. For the treatment with Endosidin-2 (ES2) (Sigma-Aldrich), HeLa cells were treated with ES2 (40 μM) for 4 hr as described in a previous report [29], after which cells were washed with warm EM buffer for 3 times prior to *S. flexneri* infection. Cells treated with 0.01% DMSO for the same incubation period served as control.

## Magnetic purification of infection-associated macropinosomes (IAMs)

We performed magnetic purification of IAMs as described previously [6]. In brief, iron oxide composite magnetic particles functionalized with dextran conjugated with Rhodamine-B based red dye DY-555 moiety (100 nm in diameter) was used (Micromod-#94-00-102-S). About 1.6 x $10^7$ HeLa cells were seeded in a T225 culture flask two days prior to infection. Cells were infected with *S. flexneri* at MOI of 50 in EM buffer in the presence of the magnetic particles (6.8 x $10^{11}$ particles/mL) for 30 min at 37 °C in the presence of 5% $CO_2$. HeLa cells only incubated with the same number of magnetic particles served as a background control. After 30-min incubation, cells were washed with ice-cold homogenization buffer (20 mM HEPES-KOH, pH 7.4, 250 mM sucrose, 0.5 mM ethylene glycol tetraacetic acid (EGTA)) supplemented with the protease inhibitor cocktail (Roche) and 5 μg/mL cytochalasin-D (Sigma-Aldrich). Cells were lyzed in homogenization buffer using a Dounce homogenizer at 4 °C. The efficiency of cell lysis was confirmed under a light microscope. Nuclei and any intact cells were removed by centrifugation twice at 200 *g* for 10 min at 4 °C, after which the homogenate fraction (H fraction) was obtained.

For magnetic purification, HOKImag free-flow magnetic separation system with a high-gradient 2-Tesla permanent magnetic field (Hoock GMBH) connected to a peristaltic pump (Gilson) was used. The H fraction was first loaded into the system. After that, the liquid flow direction was reversed, where the eluent was collected as the non-magnetic fraction (NM fraction). After washing, the plastic tubing located between the magnets was excised and the magnetic fraction (M fraction) was obtained by pushing the clumps of magnetic particles out of the tubing using a piston of the disposable syringe. Each fraction was sonicated to break the membrane of IAMs to release the magnetic particles. Magnetic particles in the M fractions were then removed by centrifugation under a sucrose gradient (45% (w/v)) at 25000 *g* for 2 hr at 4 °C. The solution on top of the sucrose layer was collected after centrifugation. Efficiency of the magnet purification were analyzed by Tecan fluorescence microplate reader using excitation filter 560±10 nm and emission filter 665±8 nm. M and NM fractions of infected and control samples were analyzed under Perkin Elmer Ultraview spinning disk confocal microscope using 60X/1.2 NA water objective after staining with membrane dye FM2-10 (Thermo Fisher). The protein concentrations were determined by BCA protein assay (Thermo Fisher). 60 μg of proteins of M and NM fraction was precipitated using trichloroacetic acid (TCA)-acetone precipitation and the pellet was stored at -20 °C until the analysis by mass spectrometry.

## Proteomic analysis

Protein samples were re-suspended in 8 M urea/ 100 mM Tris-HCl, pH 8.5. Samples were reduced using 5 mM tris(2-carboxyethyl)phosphine (TCEP) for 30 min then alkylated using

10 mM iodoacetamide for 30 min at room temperature in the dark. A first digestion was done with rLys-C Mass Spec Grade (Promega, Madison, WI, USA) in 80:1 protein-to-LysC ratio at 30 $^{\circ}$C for 3 hr. Samples were then diluted to 2 M urea with 100 mM Tris HCl, pH 8.5 and Sequencing-Grade Modified Trypsin (Promega, Madison, WI, USA) was added in 50:1 ratio for the second digestion overnight at 37 $^{\circ}$C. A second incubation with the same amount of trypsin was performed (5 hr at 37˚C) to ensure a complete digestion. Formic acid (5%) was added to quench the digestion. Resulting peptides were desalted and concentrated using Sep-Pak $C_{18}$ SPE cartridge (Waters Milford, MA, USA) according to manufacturer instructions.

Tryptic peptides were analyzed using Q Exactive Plus instrument (Thermo Fisher Scientific, Bremen) coupled with an EASY nLC 1000 chromatography system (Thermo Fisher Scientific). Sample was loaded on an in-house packed 50 cm nano-HPLC column (inner diameter of 75 μm) with a $C_{18}$ resin (particle size of 1.9 μm, pore size of 100 Å, Reprosil-Pur Basic C18-HD resin, Dr. Maisch GmbH, Ammerbuch-Entringen) and equilibrated with 98% solvent A (0.1% formic acid in water) and 2% solvent B (0.1% formic acid in acetonitrile). Samples were loaded to the column and eluted using a gradient of 5–22% gradient of solvent B in 150 min, followed by 22–45% gradient of solvent B in 60 min and a flow of 45–80% gradient of solvent B in 10 min at 250 nL/min. Ten most intense precursor ions were selected for higher-energy collisional dissociation (HCD) fragmentation with a normalized collision energy set at 28 after a survey scan in Orbitrap (resolution 70000). Precursors with unknown charge state and a charge state of 1 and >7 were eliminated. Dynamic exclusion was set to be 45s.

Triplicates of the infected and control samples were analyzed. Andromeda using MaxQuant software 1.5.3.8 [51, 52] was analyzed against Uniprot proteome database of Human (v20150113, 89706 entries), *S. flexneri* strain M90T (3819 entries) and the virulent plasmid pWR100 (104 entries). Mass tolerance was set to be 20 ppm for the first search, followed by 6 ppm and 10 ppm for the main search and for MS/MS, respectively. The minimal peptide length was set to be 5 while maximum peptide charge was set to be 7. Match between runs feature was used between conditions with a maximal retention time of 1 min. At least one unique peptide of the protein group was needed for protein identification. False discovery rate cut-off was set at 1%.

Comparative analyses of the magnetic (M) against the non-magnetic (NM) fractions of the infected sample (INF-M *vs* INF-NM) and the magnetic fractions of the infected (INF-M) against that of control sample (Ctrl-M) (INF-M *vs* Ctrl-M) were performed. Proteins with significantly differentiated abundance comparing two fractions were analyzed according to the following pipeline: (1) reverse and potential contaminants of mass spectrometry were eliminated; (2) only proteins with at least 2 quantified values were considered; (3) intensities of proteins were processed by log-2; (4) intensities were normalized by median centering within conditions using R package DAPAR [53]; (5) proteins that are present in only one fraction out of the two comparing fractions were grouped and isolated as "differentially abundant proteins"; (6) missing value of the remaining proteins was filled using impute.slsa in R package imp4p (https://cran.r-project.org/web/packages/imp4p/index.html); (7) statistical analysis was performed using a LIMMA t-test [54] coupled with an adaptive Benjamini-Hochberg correction of the p-values. False discovery rate was set to be 1% to classified the significantly differentially abundant proteins using R package cp4p [55]. The mass spectrometry proteomics data have been deposited to the ProteomeXchange Consortium via the PRIDE partner repository with the dataset identifier PXD020161. Analysis on the functional annotation clustering of the filtered results was performed by online software DAVID [56].

## Time-lapse microscopy

6000 cells per well were seeded in 4-well inserts (Ibidi-#80649) in glass-bottom dishes (Ibidi-#81158) three days prior to infection experiment. Plasmid transfection was

performed the following day using FuGENE HD transfection reagent (Promega) for 48 h. On the day of infection, cells were washed with warm EM buffer for three times. Then, cells were challenged with the bacteria at a MOI of 50. Live imaging was performed on DeltaVision Elite (GE Healthcare) using Olympus 60X/1.42 NA oil objective with refractive index oil 1.520 (GE Healthcare). Images were recorded every minute for 80 min at a step-size of 0.3 μm in z-plane at 37 °C. Images were processed using the built-in deconvolution analysis module.

## Quantitative image analysis on the distribution of the IAMs by fluorescent dextran

6000 cells per well were seeded in a 96-well plate three days prior to infection. Plasmid transfection was performed the following day using FuGENE HD transfection reagent (Promega) for 48 h. On the infection day, cells were washed three times with warm EM buffer and then challenged with *S. flexneri* at a MOI of 20. Bacteria mixed with 0.5 μg/mL Dextran-647 MW10,000 Da (Thermo Fisher). Bacteria (together with Dextran) were first incubated with HeLa cells at room condition for 10 min prior to infection at 37 °C for 30 min. After that, cells were washed 5 times with ice-cold PBS buffer and then fixed with 4% paraformaldehyde at room condition for 15 min. Cell nuclei and bacteria were stained with DAPI (1 ng/mL) (Thermo Fisher) and actin foci were labeled by Rhodamine-phalloidin (Thermo Fisher) at room condition for 20 min. Ten random images were taken per sample using a Perkin Elmer Ultraview spinning disk confocal microscope using a 60X/1.2 NA water objective at a step-size of 0.3 μm in z-plane. Quantification analysis of IAM formation at individual infection focus was performed using the software Cell Profiler as described previously [16]. The number and the sizes of the IAMs in relation to the number of bacteria at each individual infection focus were obtained. Integrated areas of the IAMs at individual infection focus were analyzed by the software Image J. Representative images illustrating the quantification procedure were shown in S4M Fig. In brief, the IAMs (stained by fluorescent dextran) in the confocal images were segmented and were processed as binary images. The adjacent IAMs were connected using the built-in "dilate" function of the binary processing in Image J, after which patches that consist of clustered IAMs were obtained. Any empty spaces within the patches were filled using the built-in "fill" function of the binary processing in Image J. The area occupied by the IAMs at an individual infection focus was estimated by measuring the area that bordered these patches at the given focus using the built-in "analyse particles" function of Image J. The distance in 3D between individual IAM and *S. flexneri* at the entry focus was analyzed using software Icy (http://icy.bioimageanalysis.org/) (S4N Fig). "HK-Means" plugin was applied to segment dextran-filled IAMs and DAPI-stained *S. flexneri* in 3D confocal stacks. At least 180 IAMs in at least 15 invasion foci from 3 independent experiments were analyzed in each condition. The x-, y-, z-coordinates of the IAM and *S. flexneri* were extracted and the distance between the IAM and *S. flexneri* was then calculated taking into consideration the pixel sizes in the x-plane (0.12 μm), y-plane (0.12 μm) and z-plane (0.3 μm) using the equation:

$$\sqrt{\left[(x_{IAM} - x_{Shigella}) \times 0.12\right]^2 + \left[(y_{IAM} - y_{Shigella}) \times 0.12\right]^2 + \left[(z_{IAM} - z_{Shigella}) \times 0.3\right]^2}$$

, where x, y, z represents the x-, y- and z-coordinates of the IAM or *S. flexneri*

The distance between IAM and *S. flexneri* was obtained in μm and was plotted as mean ± SEM.

## Immunofluorescence

Uptake of the magnetic beads during *S. flexneri* invasion was examined under a Perkin Elmer Ultraview spinning disk confocal microscope using a 60X/1.2 NA water objective at a step-size of 0.3 μm in z-plane. Infection was performed as described above. Samples were fixed using 4% paraformaldehyde at room temperature for 15 min. For immunofluorescence using Rab5A antibody, the cells were permeabilized using 0.5% saponin at room temperature for 5 min. Samples were then incubated in blocking buffer (1% bovine serum albumin, 10% fetal bovine serum in PBS buffer) at room temperature for 30 min prior to primary antibody staining using anti-mouse anti-Rab5A antibody (BD Bioscience) in 1:500 dilution. For immunofluorescence using Sec3, Sec5, Exo70 antibodies, cells were permeabilized and blocked with 0.25% saponin in the presence of 0.5% bovine serum albumin, 10% fetal bovine serum in PBS buffer for 1 hour at room temperature prior to primary antibody staining using anti-rabbit anti-Sec3 antibody (Proteintech-#11690-1-AP) in 1:100 dilution or anti-mouse anti-Sec5/Exoc2 antibody (Proteintech-#66011-1-Ig) in 1:200 dilution or anti-mouse anti-Exo70 antibody (Sigma-#WH0023265M1) used at 1:200 dilution. For immunofluorescence using Rab8A and Rab11A antibody, cells were permeabilized with 0.1% Triton-X-100 for 15 min. Samples were then incubated with blocking buffer (1% bovine serum albumin, 10% fetal bovine serum in PBS buffer) at room temperature for 1 hour prior to primary antibody staining using anti-mouse anti-Rab8A antibody (BD Bioscience) in 1:500 dilution or anti-rabbit anti-Rab11A antibody (Thermo Fisher) in 1:100 dilution. Samples were washed for three times with PBS buffer prior to secondary anti-mouse or anti-rabbit antibody conjugated to farred dye Cy5 (Thermo Fisher) in 1:500 dilution.

## Gentamicin protection assay

HeLa cells pre-treated with siRNA or pre-treated with ES2 (50 μM) were infected with *S. flexneri* as described above. Gentamicin protection assay was performed according to a previous report [17] with minor modifications. After infection, cells were incubated with EM buffer containing 100 μg/mL gentamicin for 1 hr at 37 °C with 5% $CO_2$. After a 1-hr incubation, cells were incubated with EM buffer containing 10 μg/mL gentamicin for another hour at 37 °C with 5% $CO_2$. After that, cells were washed with PBS for 3 times and were harvested using 0.05% Trypsin-EDTA (Thermo Fisher). The number of cells after infection was counted and cells were lysed by distilled water containing 0.5% Triton-X-100 at room condition for 15 min. Bacteria were serially diluted and were plated on TCS agar supplemented with 100 μg/mL ampicillin. The colony-forming unit (CFU) was determined after an overnight incubation at 37 °C. Reverse-transcription-quantitative polymerase chain reaction and Western blot were employed to confirm the efficiency of siRNA treatment. For details on the primers for reverse-transcription-quantitative polymerase chain reaction, see S3 Table.

## Western blotting

To evaluate the knock-down efficiency for siRNA experiment, cells treated with siRNA were harvested at 72-hr post-transfection. Cells were lysed with RIPA buffer (Thermo Fisher) and cell lysate was mixed with 4X Laemmli sample buffer (Bio-rad). 50 μg of cell lysate of each samples were analyzed using 10% Bis-Tris NuPAGE gels (Thermo Fisher). Immunoblotting was performed with methanol-activated PVDF membrane with the primary antibody anti-mouse monoclonal anti-Exo70 (Sigma-#WH0023265M1) used at 1:1000 dilution or anti-mouse anti-Sec5/Exoc2 (Proteintech-#66011-1-Ig) used at 1:2000 dilution and with the secondary anti-mouse antibody conjugated with horseradish peroxidase (Thermo Scientific) diluted at 1:5000.

Western blot images were finally developed using SuperSignal West Pico PLUS Chemilumi-nescent Substrate (Thermo Fisher).

## Statistical analysis

Statistical analyses were performed using software GraphPad Prism v6. Two-tail unpaired t-test was performed, where $p < 0.05$ was considered as statistically significant: $^*p < 0.05$, $^{**}p < 0.01$, $^{***}p < 0.001$, $^{****}p < 0.0001$.

## Supporting information

**S1 Table. Plasmids used in this study.**
(DOCX)

**S2 Table. Primers for cloning.**
(DOCX)

**S3 Table. Primers for real-time quantitative PCR.**
(DOCX)

**S4 Table. Some exocyst subunits and the regulatory GTPase are enriched at the IAMs (INF-M *vs* INF-NM).**
(DOCX)

**S5 Table. The exocyst subunit and the regulatory GTPase are enriched at the IAMs (INF-M *vs* Ctrl-M).**
(DOCX)

**S1 Fig. Labeling of *S. flexneri* IAMs by magnetic beads to be enriched for subsequent prote-omic analysis.** (A) Confocal image of z-projection of a representative infection focus of *S. flex-neri*-infected HeLa in the presence of magnetic beads. Inset of the infection focus was highlighted in yellow and was shown, where *S. flexneri* was stained by DAPI (blue) and actin was stained by phalloidin (green). Immunofluorescence staining of endogenous Rab5A (in grey), a macropinosome marker, indicated that magnetic beads used in this study (in red) were present in Rab5A-positive IAMs. Scale bar is 10 μm. (B) Analysis of annotated clustering was performed by online software DAVID. The enriched clusters in the respective comparison were listed according to the descending order of the enrichment score.
(TIF)

**S2 Fig. Recruitment of the exocyst to *S. flexneri* IAMs.** Confocal images of z-projection of representative infection foci of *S. flexneri*-infected HeLa cells. Insets of the infection foci were highlighted in yellow and were shown in A-C. Scale bars are 5 μm. Immunofluorescence stain-ing with GFP-tagged Rab11A confirmed that endogenous (A) Sec3 (magenta), (B) Sec5 (magenta) and (C) Exo70 (magenta) were recruited to *S. flexneri* IAMs that are labelled by Rab11A (green) (as marked by the white arrows). Nuclei and *S. flexneri* were stained by DAPI (blue). Time-lapse microscopic analysis of the recruitment of GFP-tagged (D) Sec8 and (E) Sec15 (in grey) in the presence of Galectin-3-mOrange (red), a marker of vacuolar rupture. Images were recorded every minute and z-projections of representative entry sites are shown. The white arrowheads indicate the IAMs enriched with the respective exocyst subunits. Scale bars are 5 μm.
(TIF)

**S3 Fig. The exocyst is not necessary for the early entry steps of *S. flexneri* invasion.** (A) Analysis of the timing of actin cage formation (green) and BCV rupture (dotted red ellipse)

with reference to the onset of membrane ruffling during infection foci formation (black). (B) Knock-down efficiency of Exo70 by RNA interference (siRNA Exo70) was confirmed by Western blotting, whereas non-targeting RNA (siRNA Neg) was used as a control. Actin was used as the loading control of the Western blot. Time-lapse microscopic analyses of *S. flexneri* infection of control HeLa cells and Exo70-depleted HeLa cells were performed to examine the early entry steps of *S. flexneri* invasion as illustrated in (B), including (C) time of foci formation, (D) time of actin cage formation and (E) time of BCV rupture. At least a total of 135 infection foci (n > 135), 135 ruptured BCVs (n > 135) and 160 actin cages (n > 160) in triplicate experiments were analyzed in each condition. The bars (magenta) represent the mean and the unpaired t-tests were carried out (ns: non-significant).
(TIF)

**S4 Fig. Disruption of the exocyst reduces IAMs clustering around *S. flexneri* but does not affect the formation of IAMs.** *S. flexneri* invasion of HeLa cells was performed in the presence of fluorescent dextran. The number of infection foci per HeLa cell, the area of the actin foci and the number of IAMs per *S. flexneri* were evaluated in different conditions including (A-C) RNA interference of non-targeting control (siRNA Neg) versus Exo70 depletion (siRNA Exo70), (D-F) RNA interference of non-targeting control (siRNA Neg) versus Sec5 depletion (siRNA Sec5), (G-I) expression of wild-type Sec5 or Sec5 lacking the core exocyst assembly motif (Sec5-ΔCorEx) and (J-L) expression of wild-type Exo70 or Exo70 lacking the core exocyst assembly motif (Exo70-ΔCorEx). At least 25 images consisting of more than 50 infection foci (n > 50) were analyzed in triplicates of each condition. The bars (magenta) represent the mean and unpaired t-tests were carried out (ns: non-significant). (M) Quantification of the areas occupied by IAMs per *S. flexneri* using software Image J. Confocal images of *S. flexneri* infection in the presence of fluorescent dextran (magenta) were analyzed. Nuclei and *S. flexneri* were stained by DAPI (blue) while the actin infection focus was marked by phalloidin (red). Scale bar is 5 μm. (i) Infection focus of *S. flexneri* was selected and the channel with IAM (marked by the fluorescent dextran) was segmented. (ii) Binary image of the dextran channel was obtained. (iii) Adjacent IAMs were connected to form a patch that outlined the clustered IAMs, while any empty spaces within the patch were filled. (iv) The area occupied by the IAMs at an individual infection focus was estimated by measuring the area bordering the patches of the clustered IAMs at the given focus using the built-in Analyze Particles function. (N) The spatial distribution of IAMs around *S. flexneri* at infection focus was examined in 3D using software Icy. Inset of the infection focus was highlighted in yellow. The distances between individual IAMs (in blue) and *S. flexneri* (in red) were estimated, which is marked by the yellow line. (O) Knock-down efficiency of Sec5 by RNA interference (siRNA Sec5) was confirmed by Western blotting, whereas non-targeting RNA (siRNA Neg) was used as a control. Actin was used as the loading control. (P) Distribution of the diameters of IAMs at *Shigella*-infected Sec5-depleted (siRNA Sec5) or Exo70-depleted (siRNA Exo70) or control HeLa cells (n>1500 IAMs in 3 independent experiments).
(TIF)

**S5 Fig. Disruption of the assembly of the exocyst does not affect the early entry steps of *S. flexneri* but reduces the bacterial load at later stages.** (A) Analysis of the number of bacterium per infection focus after challenging HeLa cells with *S. flexneri* for 30 min in Sec5- or Exo70-depleted cells or in control HeLa cells (n > 50). Data were shown as mean ± SEM (ns: non-significant). (B) Gentamicin protection assay of *S. flexneri* infection of Sec5- or Exo70-depleted and control HeLa cells was performed to study the effect of depleting Exo70 on the later stage of *S. flexneri* invasion (2hr-post infection). CFU in Exo70-depleted condition was normalized against that in HeLa control. Data were shown as mean ± SEM (n = 4) (**p<0.01;

*p<0.05). Time-lapse microscopy was employed to examine the time of actin foci formation, time of actin cage formation and time of BCV rupture in RNA interference of the non-targeting control (siRNA Neg) versus Sec5-depletion (C-E), cells expressing Sec5 or Sec5 lacking the core exocyst assembly motif (Sec5-ΔCorEx) (F-H) and in cells expressing Exo70 or Exo70 lacking the core exocyst assembly motif (Exo70-ΔCorEx) (I-K). Infection foci (n > 95), actin cage (n > 45) and individual BCVs (n > 65) were analyzed in triplicates for each condition. The bars (magenta) represent the mean and unpaired t-tests were carried out (ns: non-significant). (TIF)

**S6 Fig. Analysis of the effects of Endosidin-2 (ES2), the chemical inhibitor of Exo70, on *S. flexneri* infection.** Time-lapse microscopy was employed to examine (A) the time of actin foci formation, (B) time of actin cage formation and (C) time of BCV rupture in ES2-treated and DMSO control cells. *S. flexneri* infection of ES2-treated and DMSO control cells in the presence of the fluorescent dextran was fixed at 30min-post infection and (D) the area of the actin foci, (E) number of infection foci per HeLa cell, (F) the number of IAMs per *S. flexneri*, (G) the area occupied by IAMs per *S. flexneri* were evaluated. Infection foci (n > 55) were examined in triplicates for each analysis. The areas occupied by IAMs were estimated by software ImageJ as illustrated in S4M Fig while (H) the distance between individual IAM and *S. flexneri* was estimated by software Icy in 3D as illustrated in S4N Fig. (I) Gentamicin protection assay of *S. flexneri* infection of ES2-treated and control HeLa cells was performed (2 hr-post infection). CFU in ES2-treated condition was normalized against that in HeLa control. Data were shown as mean ± SEM (n = 3) (**p<0.01). (J) Analysis of the fates of individual *S. flexneri* and their BCVs with reference to the observations in Fig 4A in ES2-treated and control cells. The data are shown in mean ± SEM (n = 3) (ns: non-significant; *p<0.05). The bars (magenta) represent the mean and unpaired t-tests were carried out (****p<0.0001). Time-lapse microscopic examination of (K) the time of BCV fragmentation and (L) formation of the actin tail in ES2-treated and DMSO control. The bars (magenta) represent the mean and unpaired t-tests were carried out (ns: non-significant; ***p<0.001; ****p<0.0001). (TIF)

**S7 Fig. Rab8A and Rab11A impairment has no effect on the formation of IAMs but affects IAM clustering around *S. flexneri*.** (A) Time-lapse microscopic images of the recruitment of Rab8A (green) with reference to the BCV rupture marked by Galectin-3 (red). Images were captured every minute and the z-projections of a representative infection focus were shown. The white arrowheads indicate some of the Rab8A-positive IAMs. Scale bar is 5 μm. (B) Confocal image of z-projection of a representative infection focus of *S. flexneri*-infected HeLa. Inset of the infection focus was highlighted in yellow and was shown, where nuclei and *S. flexneri* were stained by DAPI (blue). Immunofluorescence staining of endogenous Rab8A showed that GFP-tagged Sec5 colocalized with Rab8A on some IAMs (marked by the yellow arrowheads). Scale bar is 5 μm (C) Number of infection foci per HeLa cells, (D) number of IAMs per *S. flexneri* and (E) area occupied by IAMs per *S. flexneri* at individual infection foci in Rab11A, dominant negative mutant of Rab11A (Rab11A-DN), Rab8A and dominant negative mutant of Rab8A (Rab8A-DN). 29 images consisting of at least 90 infection foci (n > 90) were analyzed in triplicate experiments of each condition. The bars (magenta) represent the mean and the unpaired t-tests were carried out (ns: non-significant; **p<0.01; ****p<0.0001). Distance between individual IAM and *S. flexneri* was estimated in 3D by software Icy as illustrated in S4N Fig and compared in (F) cells expressing Rab8A and Rab8A-DN and (G) cells expressing Rab11A and Rab11A-DN. The bars (magenta) represent the mean and unpaired t-tests were carried out (****p<0.0001). (TIF)

**S1 Movie. Connected to Fig 2A: Time-lapse microscopy shows the recruitment of Sec3 to *S. flexneri* IAMs and to close proximity to *S. flexneri* BCV.** Wild-type *S. flexneri* infection of cells transfected with GFP-Sec3 (grey) and the vacuolar rupture marker Galectin-3-mOrange (red). Sec3-positive IAMs are observed before BCV rupture. Sec3 were transiently localized at close proximity to the BCV. Images were taken every 1 min and z-projections are presented. The yellow ellipses indicate the intracellular bacteria. (MOV)

**S2 Movie. Connected to Fig 2B: Time-lapse microscopy shows the recruitment of Sec5 to *S. flexneri* IAMs and to close proximity to *S. flexneri* BCV.** Wild-type *S. flexneri* infection of cells transfected with GFP-Sec5 (grey) and the vacuolar rupture marker Galectin-3-mOrange (red). Sec5-positive IAMs are observed before BCV rupture. Sec5 were localized at close proximity to the BCV. Images were taken every 1 min and z-projections are presented. The yellow ellipses indicate the intracellular bacteria. (MOV)

**S3 Movie. Connected to Fig 2C: Time-lapse microscopy shows the recruitment of Exo70 to *S. flexneri* IAMs.** Wild-type *S. flexneri* infection of cells transfected with GFP-Exo70 (grey) and the vacuolar rupture marker Galectin-3-mOrange (red). Exo70-positive IAMs are observed at *S. flexneri* IAMs before BCV rupture. Images were taken every 1 min and z-projections are presented. The yellow ellipses indicate the intracellular bacteria. (AVI)

**S4 Movie. Connected to Fig 4A: Time-lapse microscopy demonstrates the displacement of BCV membrane remnants away from *S. flexneri* and the bacterium becomes "naked" (free from BCV fragments).** Wild-type GFP-expressing *S. flexneri* (green) infection of cells transfected with vacuolar rupture marker Galectin-3-mOrange (red). BCV fragmentation started at the moment of BCV rupture before *S. flexneri* started moving 10 min post-BCV rupture. The bacterium became naked and moved out of field in less than 20 min. Images were taken every 1 min and z-projections are shown. The instant of BCV rupture (as indicated by the appearance of the Galectin-3 signal at the BCV) was set as time-0. (MOV)

**S5 Movie. Connected to Fig 4A (middle panel–"Remnant"): Time-lapse microscopy reveals *S. flexneri* moving with the attachment of the damaged BCV.** Wild-type GFP-expressing *S. flexneri* (green) infection of cells transfected with vacuolar rupture marker Galectin-3-mOrange (red). *S. flexneri* moved with the attachment of the damaged BCV for 20 min after BCV rupture. The bacterium moved out of field at 25 min post-BCV rupture. Images were taken every 1 min and z-projections are shown. The instant of BCV rupture (as indicated by the appearance of the Galectin-3 signal at the BCV) was set as time-0. (MOV)

**S6 Movie. Connected to Fig 6A: Time-lapse microscopy shows the Rab8A-dependent recruitment of Exo70 to *S. flexneri* IAMs.** Wild-type *S. flexneri* infection of cells transfected with GFP-Exo70 (green) and Rab8A-mApple (red). Exo70 colocalized on some Rab8A-positive IAMs. Images were taken every 1 min and z-projections are presented. The white ellipses indicate the intracellular bacteria. (MOV)

**S7 Movie. Connected to Fig 6B: Rab11A is recruited to *S. flexneri* IAMs and drives the recruitment of Exo70 to *S. flexneri* IAMs.** Wild-type *S. flexneri* infection of cells transfected

with GFP-Exo70 (green) and Rab11A-mApple (red). Exo70 colocalized on some Rab11A-positive IAMs. Images were taken every 1 min and z-projections are presented. The white ellipses indicate the intracellular bacteria.
(MOV)

**S1 Appendix. Proteomes of *S. flexneri* IAMs.** Mass spectrometry-based analysis of the magnetically-purified IAMs from *S. flexneri*-infected HeLa cells. Magnetic (M) fractions and non-magnetic (NM) fractions were obtained for *S. flexneri*-infected (INF) samples and the control (Ctrl) samples. The enrichment of proteins between (i) the M fraction and NM fraction of the infected samples (INF-M *vs* INF-NM) and between (ii) the M fraction of infected samples and M fraction of uninfected control (INF-M *vs* Ctrl-M) were compared and are shown in the spreadsheets. Results on the comparison INF-M *vs* INF-NM are grouped with green tags, whereas the results on the comparison INF-M *vs* Ctrl-M are grouped with blue tags.
(XLSX)

## Acknowledgments

We acknowledge the helpful discussions with all lab members of the DIHP unit, in particular Camille Rey, Magdalena Gil Taran and Laura Barrio Cano for their feedback on the manuscript. We thank Aleix Boquet-Pujadas for his technical support in image analysis and software Icy.

## Author Contributions

**Conceptualization:** Yuen-Yan Chang, Virginie Stévenin, Jost Enninga.

**Data curation:** Yuen-Yan Chang, Magalie Duchateau, Quentin Giai Gianetto, Veronique Hourdel, Cristina Dias Rodrigues, Mariette Matondo.

**Formal analysis:** Yuen-Yan Chang, Magalie Duchateau, Quentin Giai Gianetto, Veronique Hourdel, Mariette Matondo.

**Funding acquisition:** Yuen-Yan Chang, Jost Enninga.

**Investigation:** Yuen-Yan Chang, Cristina Dias Rodrigues.

**Methodology:** Yuen-Yan Chang, Virginie Stévenin, Norbert Reiling.

**Project administration:** Yuen-Yan Chang, Jost Enninga.

**Resources:** Mariette Matondo, Norbert Reiling.

**Software:** Quentin Giai Gianetto.

**Supervision:** Mariette Matondo, Jost Enninga.

**Validation:** Yuen-Yan Chang, Cristina Dias Rodrigues.

**Visualization:** Yuen-Yan Chang.

**Writing – original draft:** Yuen-Yan Chang, Jost Enninga.

**Writing – review & editing:** Yuen-Yan Chang, Jost Enninga.

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
