## [Decision Letter · Decision Letter 0]

15 May 2020

Dear Dr. Enninga,

Thank you very much for submitting your manuscript "Shigella hijacks the exocyst to cluster macropinosomes for efficient vacuolar escape" for consideration at PLOS Pathogens. As with all papers reviewed by the journal, your manuscript was reviewed by members of the editorial board and by several independent reviewers. In light of the reviews (below this email), we would like to invite the resubmission of a significantly-revised version that takes into account the reviewers' comments.

We cannot make any decision about publication until we have seen the revised manuscript and your response to the reviewers' comments. Your revised manuscript is also likely to be sent to reviewers for further evaluation.

Sincerely,

Denise M. Monack

Section Editor

PLOS Pathogens

Denise Monack

Section Editor

PLOS Pathogens

Kasturi Haldar

Editor-in-Chief

PLOS Pathogens

orcid.org/0000-0001-5065-158X

Michael Malim

Editor-in-Chief

PLOS Pathogens

orcid.org/0000-0002-7699-2064

Reviewer's Responses to Questions

**Part I - Summary**

Reviewer #1: Entry into human epithelial cells is an essential first step in shigellosis, the diseases caused by bacteria of the genus Shigella. This intracellular pathogen induces uptake of the bacterium using a Type 3 Secretion System. Soon after entry Shigella escape from the bacteria containing vacuole (BCV) and begin actin-based motility to propel themselves through the cytosol and into neighboring cells. Recent advances in imaging methods have allowed for a more detailed understanding of this process and it is now known that Shigella entry is accompanied by newly formed vesicles (macropinosomes) that come in close proximity to the BCV. Some of the molecular players involved have been identified, largely by using a “candidate gene” approach. Biochemical experiments that could identify the identity of molecules involved in an unbiased manner have been lacking.

In this fine manuscript, Chang and co-workers use a clever approach involving magnetic beads to purify newly formed vesicles (infection associated macropinosomes-IAMs) that accompany the early steps of Shigella infection. The study begins with proteomic analysis of the purified IAMs, the authors use uninfected controls and separate their membrane fractions into magnetic and non-magnetic ones. They identify a relatively tight data set of 83 proteins. These proteins are enriched for microtubule proteins and also those involved in vesicle trafficking, especially the exocyst complex. The role of the exocyst is then explored.

Live cell imaging shows that exocyst components are recruited to IAMs around the bacteria and that Sec3 and Sec5 are present right around the BCV in the moments corresponding to vacuole rupture while the exocyst component Exo70 is restricted to IAMs. Knockdown experiments that deplete Eco70 or Sec5 show that these factors are not required for IAM formation, but they act to concentrate these vesicles around the BCV. Moreover, this clustering requires the core exocyst assembly motif that is present in 2 of the proteins of interest.

Gentimicin-protection assays performed on Sec5 and Exo70 depleted cells suggest a role of exocyst in later stages of infection and this promts the authors to carefully examine the fate of Shigella after vacuole rupture. They describe two phenotypes. The first are “naked” bacteria that rapidly break free from the BCV, which disappears and then begin actin based motility. The second population are those that remain associated with the BCV and are delayed in getting their actin based motility going. Altogether these data make a strong case for the role of exocyst in helping free Shigella to the cytosol and precisely identify the step where this occurs. Efforts are made to determine how the exocyst is recruited to the IAMs. Rab8, Rab 11, and the Shigella effector IpgD are all shown to contribute to exocyst recruitment.

This study represents an important advance in the early steps of Shigella infection. The identification of novel host factors involved in this process provides new avenues to explore.

Reviewer #2: The manuscript of Chang et al. investigates the early phase of Shigella entry into epithelial cells, the transient formation of a bacteria-containing vacuole (BCV), and the events leading of rupture of this compartment and cytosolic presence of Shigella. The Enninga group showed in early publications that interaction of the BCV with infection-associated macropinosomes (IAM) are affecting integrity of Shigella- or Salmonella-containing compartments. The present work follows a proteomic inventory of early BCV in Shigella-infected cells. An enrichment of subunits of the exocyst complex on these compartments was observed. The study further investigates the involvement of exocyst interaction of BCV and IAM by interference with exocyst subunits and specific Rab GATPases. On the bacterial side, a role of T3SS effector protein IpgD was demonstrated.

The study put focus on a very challenging event in host pathogen interaction, i.e. the transient and highly dynamic interaction of a BCV with host cell compartments between invasion and release into the cytosol. The study addresses this short and import phase by a combination of subcellular fractionation, proteomics and quantitative image analyses of live cell imaging. In addition to the technical advances provided by the work, the results provide new insight into the Shigella virulence mechanism. A novel contribution of the exocyst complex in biogenesis and rupture of BCV is provided.

The study is of high quality and provides important new results. A number of experimental issues needs further attention. The presentation of the data can also be improved.

Reviewer #3: In the manuscript entitled `Shigella hijacks the exocyst to cluster macropinosomes for efficient lysosomal escape´ by Chang et al. the authors analysed the role of the cellular exocyst for Shigella infections. The authors used a proteomic approach to characterize magnetically isolated infection-associated macropinosomes (IAMs) from Shigella-infected and uninfected cells. In IAMs fraction of infected HeLa cells, proteins of the cellular exocyst including exo1, 2, 4 and 6 were significantly enriched. Next, the author applied time-lapse microscopy to analyse the localization of ectopically expressed GFP-tagged Sec3, Sec5 and Exo70 during early infection. Then the authors applied three different approaches to address the role of Sec5 and Exo70 during infections. Firstly, they used RNA interference to silence expression of these proteins. Secondly, they ectopically expressed mutated versions of Sec5 and Exo70 that fail to assemble the exocyst and thirdly, the added an inhibitor for Exo70. By these different approaches, the authors suggest novel functions of Sec5 and Exo70 for clustering of IAMs close to the bacterial containing vacuole (BCV) and for BCV disassembly that seem to help the bacterium to escape into the cytosol, a step that is essential for bacterial propagation and spread.

The manuscript is well written and presents a new approach to study IAMs in Shigella infections and suggest a novel function of the exosome for Shigella infections.

The main problem I had with the manuscript is that ectopically over-expressed tagged exocyst subunits have been shown to mislocalize in eukaryotic cells (Matern et al., 2001; Rivers-Molina et al., 2013; Ahmed et al., 2018). Thus, conclusions based on this method (e.g. Figure 1 and Figure 6) could be misleading.

**Part II – Major Issues: Key Experiments Required for Acceptance**

Reviewer #1: I require no additional experiments.

Reviewer #2: A key message of the manuscript is the formation of two distinct compartments during Shigella invasion, i.e. the BCV containing Shigella, and the IAM formed as distinct entities. The infection protocol applied is critical, since magnetic beads for labeling of IAM and Shigella for invasion and formation of BCV were applied at the same time. Given the massive membrane remodelling during Shigella-mediated trigger invasion, it is rather likely that bacteria and magnetic beads are co-internalized in the same compartment. Please provide controls that distinct IAM and PCV exist prior to interaction of these entities and the rupture of the BCV. If IAM are only formed during infection, designing a clean control will be a tough cookie. As temporal dissection will not be possible, spatial separation by microscopy remains as option.

Along these lines, it should be considered if HeLa cells with magnetic bead only is the most suitable control. Without any stimulus for macropinocytosis, this control will be ‘super-negative’, since no macropinosomes are formed, and enrichment of INF-M over Ctrl-M has to be immense. Consider to apply an invasion-independent stimulus for macropinocytosis, followed by isolation of the resulting compartments as control for proteomics.

Quantitative data (x-fold values) should be provided for the enrichment shown in Fig. 1D. The analysis was defined as quantitative proteomics, and the enrichment factors should be given for the hits that were followed up.

The quantitative image analysis workflow described in Fig. S4M is critical for the understanding of the results. However, the representation of the workflow is not fully intuitive and should be explained in more detail.

Reviewer #3: (1) Ectopic expression: Exocyst subunits were systematically tagged with GFP in MDCK cells a decade ago, but most GFP fusions produced diffuse cytosolic labeling, incongruent with localization to vesicles (Matern et al., 2001; Rivers-Molina et al., 2013; Ahmed et al., 2018). The author must overcome this obstacle. One possibility to avoid this problem was published by Rivers-Molina et al., 2013. In this paper, Sec8 subunit was silenced by RNA interference and replaced with a tagged Sec8 version. The authors must perform new experiments as outlined above to analyse the localization of the exocyst subunits and the colocalization of exocyst subunits with Rab GTPases during infection.

(2) The authors identified several proteins enriched in the IAMs fraction from infected cells. The author must validate these enriched proteins using a different method e.g staining of isolated IAMs with antibodies specific for EXOC1, EXOC2, EXOC 4, EXOC6, RAB8A. IAMs isolated from mock-infected cells should be used as controls.

(3) Sec5 and Exo70 localize to different structure during infection. Do the authors have evidence that the exocyst is formed during infection? Could both subunits act independently and control different processes? Some more data on this is needed to better understand the role of the exocyst during Shigella infection.

**Part III – Minor Issues: Editorial and Data Presentation Modifications**

Reviewer #1: Minor concerns.

The text that explains the “naked” bacteria phenotype is very difficult to read. Could the authors try to make this more plain? It may help to name the non-naked bacteria phenotype (“stuck” “trapped”?).

The explanation of the exocyst recruitment to IAMs is fair. It is difficult to imagine these data getting more conclusive since the phenotype appears to involve both host and bacteria factors that are all pleitropic in nature.

The authors may consider citing the other Shigella-GBP articles that were published in the same days around their reference (#39). They may also wish to discuss the early studies describing IpgD function and its role in “tethering” Niegbur , how does this new work advance our understanding of the role of IpgD in entry (“entry” here entry used in the broad sense).

Reviewer #2: Fig. S2AB, Fig. 2: what defines the entities highlighted by arrows as IMP? The authors used fluorescent dextran as label for live cell imaging, but a signal indicative for IMP is not registered or displayed.

The siRNA knock-down experiments shown in Fig. 3 and elsewhere would benefit from controls that report the overall effect on the compartment size of BCV and IAM.

If bacterial mutant strains are used (e.g. delta ipgD), please provide evidence that the mutation can be complemented, at least by citing prior work with such strain (basic microbiology). Is the delta ipgD strain also expression afaI?

Page 20: adhesin afaI, not adhesion afaI

Page 9 and elsewhere: Western blot, not western blot

Please provide information with isoforms of Rab GTPase was analysed. In some instances, Rab8A or Rab11A are specified, but mostly only Rab5, Rab8, or Rab11 is stated.

Fig. 5A is not intuitive. The orange box incorporates events that are much prior to actin tail formation.

In the discussion, the interactions of various intracellular pathogens with the exocytic pathway are described. The authors may wish to extend to initial and recent reports on these interactions, for example for Salmonella enterica.

Reviewer #3: (1) Abstract: the authors stated that `proteins […] were significantly enriched at the IAMs […]´. What is the evidence for this? It appears that these proteins are enriched in the fraction of IAMs isolated from infected cells. Please clarify.

(2) Are magnetic beats in uninfected cells as efficiently taken up as in infected cells?

(3) Why did the authors focus on Rab8A while Rab9 was significantly enriched in INF-M factions?

(4) The author found Exo70 in IAMs by microscopy. Why has Exo70 not been identified in the proteomic fraction?

(5) Please confirm knockdown efficiency by western blotting.

(6) How did the authors define close proximity?

(7) What is the fate of entrapped bacteria in Exo70 and Sec5 silenced cells?

(8) Does the inhibitor ES2 affect bacterial propagation?

(9) There are few sentences that are difficult to understand e.g. page 9, 1st paragraph, last sentence. Please re-write.

PLOS authors have the option to publish the peer review history of their article (what does this mean?). If published, this will include your full peer review and any attached files.

Reviewer #1: No

Reviewer #2: No

Reviewer #3: No
---

## [Editor Report · Decision Letter 1]

20 Jul 2020

Dear Dr. Enninga,

We are pleased to inform you that your manuscript 'Shigella hijacks the exocyst to cluster macropinosomes for efficient vacuolar escape' has been provisionally accepted for publication in PLOS Pathogens.

Best regards,

Denise M. Monack

Section Editor

PLOS Pathogens

Denise Monack

Section Editor

PLOS Pathogens

Kasturi Haldar

Editor-in-Chief

PLOS Pathogens

orcid.org/0000-0001-5065-158X

Michael Malim

Editor-in-Chief

PLOS Pathogens

orcid.org/0000-0002-7699-2064
---

## [Editor Report · Acceptance letter]

25 Aug 2020

Dear Dr. Enninga,

We are delighted to inform you that your manuscript, "*Shigella* hijacks the exocyst to cluster macropinosomes for efficient vacuolar escape," has been formally accepted for publication in PLOS Pathogens.

Best regards,

Kasturi Haldar

Editor-in-Chief

PLOS Pathogens

orcid.org/0000-0001-5065-158X

Michael Malim

Editor-in-Chief

PLOS Pathogens

orcid.org/0000-0002-7699-2064